# Neural Functional Transformers

**Allan Zhou**[1]    **Kaien Yang**[1]    **Yiding Jiang**[2]    **Kaylee Burns**[1]
**Winnie Xu**[1]    **Samuel Sokota**[2]    **J. Zico Kolter**[2]    **Chelsea Finn**[1]
[1]Stanford University    [2]Carnegie Mellon University
ayz@cs.stanford.edu

## Abstract

The recent success of neural networks as implicit representation of data has driven growing interest in *neural functionals*: models that can process other neural networks as input by operating directly over their weight spaces. Nevertheless, constructing expressive and efficient neural functional architectures that can handle high-dimensional weight-space objects remains challenging. This paper uses the attention mechanism to define a novel set of permutation equivariant weight-space layers and composes them into deep equivariant models called *neural functional Transformers* (NFTs). NFTs respect weight-space permutation symmetries while incorporating the advantages of attention, which have exhibited remarkable success across multiple domains. In experiments processing the weights of feedforward MLPs and CNNs, we find that NFTs match or exceed the performance of prior weight-space methods. We also leverage NFTs to develop INR2ARRAY, a novel method for computing permutation invariant latent representations from the weights of implicit neural representations (INRs). Our proposed method improves INR classification accuracy by up to $+17\%$ over existing methods. We provide an implementation of our layers at `https://github.com/AllanYangZhou/nfn`.

## 1 Introduction

Deep neural networks have emerged as flexible modeling tools applicable to a wide variety of different fields, ranging from natural language processing to vision to the natural sciences. The sub-field of implicit neural representations (INRs) has achieved significant success in using neural networks to represent data such as 3D surfaces or scenes [32, 30, 37]. This has fueled interest in techniques that directly operate in weight space to modify or extract information from a neural network through its weights or gradients. However, developing models that can process weight-space objects is challenging due to their high dimensional nature. As a consequence, some existing methods for processing datasets of weights assume a restricted training process that reduces the effective weight space [9, 3, 6].

In contrast, we follow recent work in building permutation equivariant weight-space models called *neural functionals*, that can process neural network weights[1] without such restrictions [31, 46]. In particular, this work concerns neural functionals that are equivariant to permutations of the weights that correspond to re-arranging neurons. These weight-space permutations, known as *neuron permutation symmetries*, exactly preserve the network's behavior (see examples in Figure 1). Since equivariance to neuron permutations is typically a useful inductive bias for weight-space tasks, neural functionals achieve superior generalization compared to non-equivariant architectures. Despite this, their performance on weight-space tasks remains significantly worse than convolutional networks' performance on analogous image-space tasks [31, 46], suggesting that neural functional architectures can be significantly improved.

---

[1]For clarity, we use "weights" (and "weight space") to describe the weights (and space they belong to) for the network being processed by a neural functional. We use "parameters" for the weights of the neural functional.

37th Conference on Neural Information Processing Systems (NeurIPS 2023).

While existing neural functionals rely on *linear* layers (analogous to convolution), some of the most successful architectures in other domains, such as Transformers [41, 8], rely on *nonlinear* attention mechanisms. Motivated by this fact, our work develops novel equivariant weight-space layers based on attention. While naive self-attention between input weights is permutation equivariant, it cannot distinguish true neuron permutation symmetries (which preserve network behavior) from "false" permutation symmetries[2] (which do not preserve network behavior). In contrast, we prove that our weight-space self-attention layer is *minimally equivariant*: it is equivariant to *only* neuron permutations, which are the actual weight-space symmetries.

When composed into deep architectures, our weight-space attention layers give rise to neural functional transformers (NFTs). As an immediate application, we use NFTs to develop INR2ARRAY, a method for mapping INR weights into compact latent representations that are trained through a reconstruction-based objective. INR2ARRAY produces permutation-invariant latents that can be useful for downstream tasks such as INR classification. We also construct NFTs for a variety of other tasks such as weight-space editing or generalization prediction.

NFTs are competitive with or outperform existing methods on each task without using more parameters. Notably, NFTs and INR2ARRAY improve downstream INR classification accuracy over the best existing methods on multiple datasets, by up to $+17\%$. Overall, our experiments show that attention-based neural functionals can be more expressive and powerful than existing architectures that rely on linear weight-space layers, leading to improved performance across weight-space tasks.

## 2  Preliminaries

Consider a feedforward network having $n_i$ neurons at each layer $i = 0, \cdots, L$, including the input and output layers. The network has weights[3] $W = \left\{ W^{(i)} \in \mathbb{R}^{n_i \times n_{i-1}} \mid i \in [\![1..L]\!] \right\}$ belonging to weight space, $\mathcal{W}$. More generally, we are interested in multi-channel weight space features that can arise from stacking multiple weight space objects such as weights and gradients. For $c$-channel weight-space features $W \in \mathcal{W}^c$, we have $W = \left\{ W^{(i)} \in \mathbb{R}^{n_i \times n_{i-1} \times c} \mid i \in [\![1..L]\!] \right\}$ and each $W^{(i)}_{jk} := W^{(i)}_{j,k,:} \in \mathbb{R}^c$ is a vector rather than a scalar.

Since the neurons in each hidden layer $i \in \{1, \cdots, L-1\}$ have no inherent ordering, the feedforward network is invariant to the symmetric group $S_{n_i}$ of permutations of the neurons in layer $i$. We follow Zhou et al. [46] in studying the **neuron permutation** (NP) group $\mathcal{S}_{\mathrm{NP}} := S_0 \times \cdots \times S_{n_L}$, which further assumes permutation symmetry of the input and output layers. Although this simplifying assumption is usually too strong, it can be effectively corrected in practice using position encodings at the input and output layers [46].

Consider a neuron permutation $\sigma = (\sigma_0, \cdots, \sigma_L) \in \mathcal{S}_{\mathrm{NP}}$. Each $\sigma_i$ permutes the neurons of layer $i$, which correspond to the output space (rows) of $W^{(i-1)}$ and the input space (columns) of $W^{(i)}$. Hence the action on weight space features $W \in \mathcal{W}^c$ is denoted $\sigma W$, where

$$[\sigma W]^{(i)}_{jk} = W^{(i)}_{\sigma_i^{-1}(j), \sigma_{i-1}^{-1}(k)}, \quad \sigma = (\sigma_0, \cdots, \sigma_L) \in \mathcal{S}_{\mathrm{NP}}. \tag{1}$$

As illustrated in Figure 1, a neuron permutation must always permute the columns of $W^{(i)}$ and the rows of $W^{(i-1)}$ simultaneously in order to preserve network behavior.

This work is focused on principled architecture design for neural functionals, i.e., architectures that process weight space features [46]. Recent work has shown the benefit of incorporating neuron permutation symmetries into neural functional architectures by enforcing equivariance (or invariance) to neuron permutation symmetries [31, 46]. Consider weight-space function classes parameterized by $\theta \in \Theta$. We say that a function class $f : \mathcal{W}^c \times \Theta \to \mathcal{W}^c$ is $\mathcal{S}_{\mathrm{NP}}$-**equivariant** if permuting the input by any $\sigma \in \mathcal{S}_{\mathrm{NP}}$ always has the same effect as permuting the output by $\sigma$:

$$\sigma f(W; \theta) = f(\sigma W; \theta), \quad \forall \sigma \in \mathcal{S}_{\mathrm{NP}}, W \in \mathcal{W}^c, \theta \in \Theta. \tag{2}$$

Similarly, a function class $f : \mathcal{W}^c \times \Theta \to \mathbb{R}$ is $\mathcal{S}_{\mathrm{NP}}$-**invariant** if $f(\sigma W; \theta) = f(W; \theta)$ for all $\sigma, W$, and $\theta$.

---

[2]Moreover, naive self-attention between input weights is computationally intractable.

[3]For simplicity, we omit the biases from our discussion here, see the Appendix for full detail.

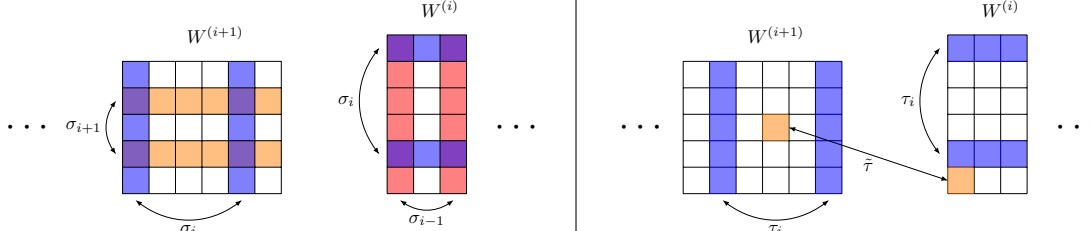

Figure 1: **Left:** An illustration of how neuron permutations $\sigma = (\sigma_0, \cdots, \sigma_L)$ act on weight matrices $W = (W^{(0)}, \cdots, W^{(L)})$. Each $\sigma_i$ simultaneously permutes the rows and columns of adjacent weight matrices $W^{(i)}, W^{(i+1)}$. **Right:** Two examples of false symmetries, i.e., permutations that don't preserve network behavior: (1) when $\tau_i$ permutes the rows and columns of adjacent matrices permute differently and (2) when $\tilde{\tau}$ permutes weights across layers.

Our weight-space layers build off of the dot-product attention mechanism, which takes a query $q$ and a set of $N$ key-value pairs $\{ (k_p, v_p) \}_{p=1}^N$ and computes, in the single-headed case:

$$\text{ATTN}\left(q, \{ (k_p, v_p) \}_{p=1}^N \right) = \sum_p v_p \left( \frac{\exp(q \cdot k_p)}{\sum_{p'} \exp(q \cdot k_{p'})} \right). \tag{3}$$

If all $q, k_p, v_p$ are vectors in $\mathbb{R}^d$, then the output is also a vector in $\mathbb{R}^d$. The definition extends to the situation where $q, k, v_p$ are multi-dimensional arrays of equal shape, by taking the dot product between these arrays in the natural way. We can also use multi-headed attention ([41]) without significant modification.

## 3 Neural Functional Transformers

We now introduce two attention-based layers for processing weight-space features: weight-space self-attention (which is $\mathcal{S}_{\text{NP}}$-*equivariant*) and weight-space cross-attention (which is $\mathcal{S}_{\text{NP}}$-*invariant*). We then describe how to compose these layers into deep architectures called neural functional transformers (NFTs).

### 3.1 Equivariant weight-space self-attention

A key concept motivating the design of our weight-space self-attention layer is the distinction between actual NP symmetries $\mathcal{S}_{\text{NP}}$ that preserve network behavior, and other permutations of the weights that can generally change network behavior. For example, consider a $\tau \in S_{\dim(\mathcal{W})}$ that permutes the columns of $W^{(i)}$ differently from the rows of $W^{(i-1)}$ or that moves weights between different weight matrices (see Figure 1). Such permutations of the weight space are called "false symmetries" [31] since they generally modify network behavior. We are interested in equivariance to the actual weight-space symmetries (neuron permutations) but *not* to any false symmetries, a property we call *minimal* equivariance:

**Definition 1.** *A function class* $f : \mathcal{W}^c \times \Theta \to \mathcal{W}^c$ *is minimally* $\mathcal{S}_{NP}$-*equivariant if it is* $\mathcal{S}_{NP}$-*equivariant (Eq. 2), but not equivariant to any false symmetries. More precisely, for any* $\tau \in S_{\dim(\mathcal{W})}$ *such that* $\tau \notin \mathcal{S}_{NP}$, *there exist a* $W \in \mathcal{W}^c$ *and* $\theta \in \Theta$ *such that* $f(\tau W; \theta) \neq \tau f(W; \theta)$.

Consider naive self-attention between the $\dim(\mathcal{W})$ weights $\{ W^{(i)}_{jk} \}$, which would be equivariant to *any* weight-space permutation. This would be $\mathcal{S}_{\text{NP}}$-equivariant, but would also be equivariant to any false permutation symmetries, meaning that it is not minimally equivariant. In other words, naive self-attention is overly constrained by too many symmetries, including those that are not actually relevant to weight-space tasks.

Our weight-space self-attention layer $\text{SA}(\cdot; \theta_Q, \theta_K, \theta_V) : \mathcal{W}^c \to \mathcal{W}^c$ is more sensitive to the distinction between neuron permutations and false symmetries. It operates on weight-space features with $c$ channels and is parameterized by query/key/value projection matrices $\theta_Q, \theta_K, \theta_V \in \mathbb{R}^{c \times c}$.

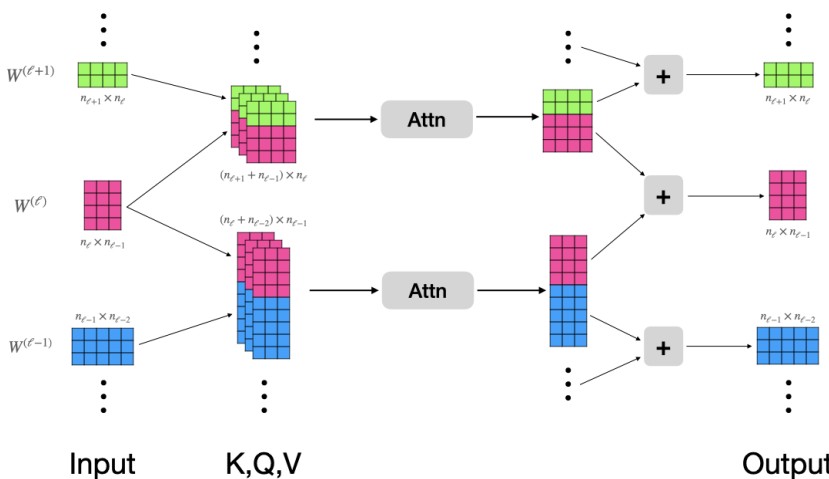

Input      K,Q,V                Output

Figure 2: An illustration of Eqs 4-5 of our weight-space self-attention layer, in the single channel case. The inputs to ATTN (Eq 3) are matrices with rows and columns corresponding to the sequence and feature dimensions, respectively. The final term (Eq 6, not shown here) simply computes self attention between all weights.

Each entry of the output feature is computed:

$$\text{SA}\left(W; \theta_Q, \theta_K, \theta_V\right)_{jk}^{(i)} = \text{ATTN}\left(Q_{j,:}^{(i)}, \left\{(K,V)_{:,q}^{(i-1)}\right\}_{q=1}^{n_{i-2}} \bigcup \left\{(K,V)_{p,:}^{(i)}\right\}_{p=1}^{n_i}\right)_k \quad (4)$$

$$+ \text{ATTN}\left(Q_{:,k}^{(i)}, \left\{(K,V)_{:,q}^{(i)}\right\}_{q=1}^{n_{i-1}} \bigcup \left\{(K,V)_{p,:}^{(i+1)}\right\}_{p=1}^{n_{i+1}}\right)_j \quad (5)$$

$$+ \text{ATTN}\left(Q_{jk}^{(i)}, \left\{(K,V)_{pq}^{(s)} \;\middle|\; \forall s,p,q\right\}\right), \text{ where} \quad (6)$$

$$Q_{j,k}^{(i)} := \theta_Q W_{jk}^{(i)}, \quad K_{j,k}^{(i)} := \theta_K W_{jk}^{(i)}, \quad V_{j,k}^{(i)} := \theta_V W_{jk}^{(i)}, \quad (7)$$

and the notation $(K,V)_{jk}^{(i)}$ is shorthand for the tuple $\left(K_{jk}^{(i)}, V_{jk}^{(i)}\right)$. Appendix A.2 defines the more general version of this layer, which handles inputs with both weights and biases.

The final term (Eq. 6) is simply naive self-attention between the inputs $\{W_{jk}^{(i)} \in \mathbb{R}^c\}$, but the first two terms give the layer additional structure relevant to neuron permutations in particular, as illustrated by Figure 2. To understand the symmetry properties of the first two terms, consider the top input to ATTN in Figure 2, which is constructed from the rows of $W^{(\ell)}$ and the columns of $W^{(ell+1)}$. If the rows of $W^{(\ell)}$ and the columns of $W^{(\ell+1)}$ are permuted simultaneously, the pairwise dot products between any two vectors in the input is preserved and the attention matrix is unchanged. Then the output of ATTN is simply permuted, giving equivariance. On the other hand, independently permuting the rows and columns of $W^{(\ell)}$ and $W^{(\ell+1)}$ will in general change the attention matrix, breaking equivariance. Since such independent permutations are false symmetries, this behavior is exactly what we need to achieve minimal equivariance.

Since neuron permutations never permute weights *between* different layers $W^{(i)}$ and $W^{(i')}$, we may additionally use learned layer position encodings $\{\phi^{(i)} \in \mathbb{R}^c \mid i \in [\![1..L]\!]\}$ to prevent equivariance to such permutations:

$$\text{LAYERENC}\left(W; \{\phi^{(i)}\}_{i=1}^L\right)_{jk}^{(i)} = W_{jk}^{(i)} + \phi^{(i)}. \quad (8)$$

We can now compose self-attention with position encoding to obtain SA ∘ LAYERENC.

**Theorem 3.1** (Minimal equivariance). *The combined layer* SA ∘ LAYERENC *is **minimally** $\mathcal{S}_{NP}$-equivariant.*

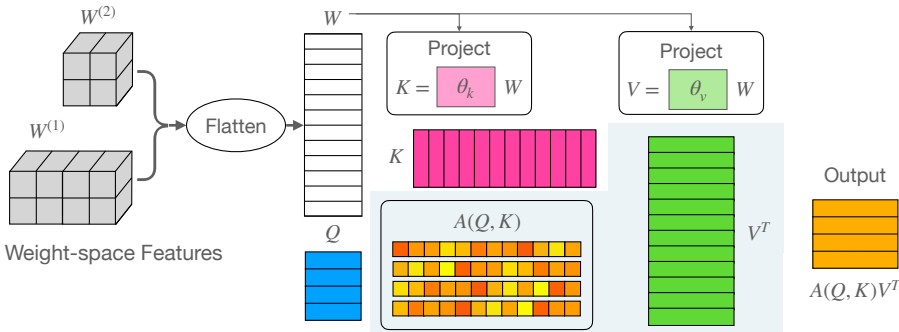

Figure 3: An illustration of our weight-space cross-attention layer, which pools weight-space features into a set of $\mathcal{S}_{\text{NP}}$-invariant vectors. The layer uses a set of learned queries $Q \in \mathbb{R}^{M \times c}$ to attend over keys and values produced from the weight space features.

*Proof (Sketch).* To show equivariance: LAYERENC is clearly equivariant to neuron permutations. We can check $\text{SA}(\sigma W)_{jk}^{(i)} = \text{SA}(W)_{\sigma_i^{-1}(j), \sigma_{i-1}^{-1}(k)}^{(i)}$ by expanding the left hand side term-by-term using the definition (Eq. 4). To show non-equivariance to false symmetries: we can broadly classify false symmetries into three types and check non-equivariance to each of them. Appendix A.1 provides the full proof. $\square$

**Practical considerations.** The final term of SA (Eq. 6) amounts to self-attention between each weight-space feature $W_{jk}^{(i)} \in \mathbb{R}^c$ and requires $O\left((\dim \mathcal{W})^2 c\right)$ operations, which is usually intractable. In practice, we replace this term by a self-attention between the row-column sums $W_{\star,\star}^{(i)} = \sum_{j,k} W_{jk}^{(i)}$ of each weight matrix. This preserves interaction between weight-space features in different weight matrices, while reducing the computational complexity to $O\left(L^2 c\right)$. One can verify that this simplified version also maintains minimal $\mathcal{S}_{\text{NP}}$-equivariance. Meanwhile, consider the self-attention operation required to compute the first two terms of SA (depicted in Figure 2). If $n_i = n$ for all $i$, then the first two terms require $O(Ln^3 c)$ operations, which is feasible for moderate $n$ but can become expensive when processing very wide networks. As a result, weight-space attention layers are usually more computationally expensive than linear weight-space layers.

### 3.2 Invariant weight-space cross-attention

Stacking weight-space self-attention layers produces equivariant weight-space features, but some situations require producing an $\mathcal{S}_{\text{NP}}$-*invariant* output. Existing approaches achieve invariance by relying on a summation or maximization over the rows and columns of each input $W^{(i)}$ [31, 46].

It is natural to extend these existing invariant layers by using attention over the input's entries, which has similar permutation invariant properties as summation and maximization. Our weight-space cross-attention layer $\text{CA} : \mathcal{W}^c \to \mathbb{R}^d$ is simply a cross attention between a learned query $e \in \mathbb{R}^d$ and the set keys and values produced from weight-space features $\{ W_{jk}^{(i)} \in \mathbb{R}^c \}$:

$$\text{CA}\left(W; e, \theta_K, \theta_V\right) = \text{ATTN}\left(e, \left\{ \left(\theta_K W_{pq}^{(s)}, \theta_V W_{pq}^{(s)}\right) \mid \forall s, p, q \right\}\right), \quad (9)$$

with $\theta_K, \theta_V \in \mathbb{R}^{d \times c}$ being learned projection matrices. By repeating this operation with $M$ different learned embeddings $e_1, \cdots, e_M$, we can easily extend this into an invariant map $\mathcal{W}^c \to \mathbb{R}^{M \times d}$. We depict the operation of the multi-embedding case in Figure 3.

### 3.3 Convolutional weight spaces

Although our description thus far focuses on fully-connected weight-space features $W$, we can also extend our layers to convolutional weight spaces. Suppose $W$ is the $c$-channel weight-space feature corresponding to a 1D convolutional network with filter widths $k_i$ at each layer $i$. It contains matrices

$W^{(i)} \in \mathbb{R}^{n_i \times n_{i-1} \times k_i \times c}$. As in Zhou et al. [46], the filter dimension $k_i$ can be folded into the channel dimension, creating a feature in $\mathbb{R}^{n_i \times n_{i-1} \times (k_i c)}$ with $k_i c$ channels. The problem is that $k_i$ may not be consistent across all $i$, while dot-product attention operations require that the channel dimensions for each $W^{(i)}$ match. We solve this problem by choosing a shared channel size $\tilde{c}$ and using learned linear projections $\text{PROJ}_i : \mathbb{R}^{n_i \times n_{i-1} \times (k_i c)} \to \mathbb{R}^{n_i \times n_{i-1} \times \tilde{c}}$ to guarantee matching channel dimensions. The output of $\text{SA}(\cdot)$ can then be restored to the original channel dimension by another set of learned linear projections $\text{UNPROJ}_i : \mathbb{R}^{n_i \times n_{i-1} \times \tilde{c}} \to \mathbb{R}^{n_i \times n_{i-1} \times k_i \times c}$.

## 3.4 Building Neural Functional Transformers

Following Transformer architecture design [41], we form stackable "blocks" that combine weight-space self attention with LayerNorm, pointwise MLPs, and residual connections. Each block maps $\mathcal{W}^c \to \mathcal{W}^c$ and preserves $\mathcal{S}_{\text{NP}}$-equivariance:

$$\text{BLOCK}(W) = Z + \text{MLP}(\text{LN}(Z)) \tag{10}$$
$$Z = W + \text{SA}(\text{LN}(W)), \tag{11}$$

where both the MLP $: \mathbb{R}^c \to \mathbb{R}^c$ and LayerNorm LN $: \mathbb{R}^c \to \mathbb{R}^c$ operate "pointwise," or independently on each input $W_{jk}^{(i)}$. Since pointwise operations are permutation equivariant, the overall block is an $\mathcal{S}_{\text{NP}}$-equivariant map on $\mathcal{W}^c$. To build neural functional Transformers (NFTs), we stack multiple blocks into a deep equivariant architecture. For tasks which require $\mathcal{S}_{\text{NP}}$-invariance, like classification or learning an invariant latent space of weights (Section 4), we can apply weight-space cross-attention on top of the features produced by the final $\text{BLOCK}(\cdot)$. The invariant activations produced by $\text{CA}(\cdot)$ can then be fed into an MLP that produces the final output.

Additionally, the NFT's input is often one or a few channels, while our desired hidden channel dimension may be significantly larger (e.g., $c = 256$). We can apply any function $g : \mathbb{R}^{c_1} \to \mathbb{R}^{c_2}$ to each $W_{jk}^{(i)} \in \mathbb{R}^{c_1}$ to project from the input channel dimension to the hidden channel dimension while preserving equivariance. In our experiments, we use random Fourier features [35] to achieve this projection.

## 4 INR2ARRAY: Learning an invariant latent space for weights

An application of interest for weight-space architectures is to learn a compact latent representation of weights, which can be useful for downstream tasks such as classifying the signal represented by an implicit neural representation (INR). The recently introduced INR2VEC [6] can learn a representation of INR weights by using a reconstruction objective, but requires every INR share the same initialization. In this section, we leverage neural functionals to extend the INR2VEC framework to the independent initialization setting, which is significantly more challenging but removes any restrictions on the INR training process.

INR2VEC uses an encoder-decoder setup to map INR weights into useful latent representations. We make two key changes to the original INR2VEC formulation:

1. We implement the encoder with an $\mathcal{S}_{\text{NP}}$-invariant neural functional transformer (NFT). This guarantees that the latent representation is invariant to neuron permutations of the weights.
2. We allow the latent representation to be a spatially meaningful *array* of vectors. In particular, each vector in the latent array is responsible for encoding a single spatial patch of the INR.

The first modification is a useful inductive bias because the signal represented by an INR should be invariant to neuron permutations. The second modification is inspired by Spatial Functa [3], which found that giving INR latent spaces spatial structure was helpful in their meta-learning setting. We call our representation learning approach INR2ARRAY due to its modified latent space structure.

The following explanation focuses on 2D INRs that represent images on the coordinate grid $[-1, 1]^2$, though the general case follows naturally. We first split the coordinate grid into $M$ spatial patches $P_1 \cup \cdots \cup P_M = [-1, 1]^2$. Given a $\text{SIREN}(\cdot; W)$, INR2ARRAY uses an $\mathcal{S}_{\text{NP}}$-invariant NFT encoder $\text{ENC}_\theta(\cdot)$ to map weights $W$ to a latent array of $M$ vectors, $z \in \mathbb{R}^{M \times d}$. The decoder $\text{DEC}_\theta : \mathbb{R}^d \to \mathcal{W}$ produces a set of weights $\left\{ \hat{W}_i \mid i = 1, \cdots, M \right\}$, one for each vector $z_i := z_{i,:} \in \mathbb{R}^d$. Each $\hat{W}_i(z_i)$ is responsible for parameterizing the SIREN only for the spatial patch $P_i \subset [-1, 1]^2$. The objective

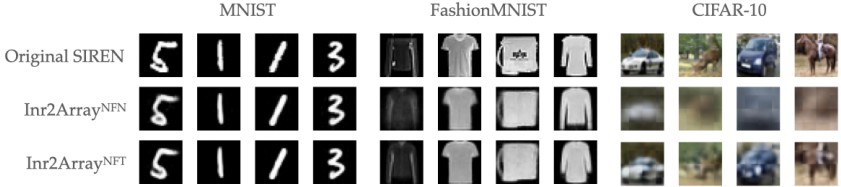

Figure 4: INR2ARRAY reconstructions for random samples from each INR dataset. The NFT encoder produces higher qualitatively better reconstructed SIRENs on each dataset, especially CIFAR-10. Blocking artifacts are due to the spatial latent structure, and suggest further room for improvement.

is to reconstruct the original INR's content using the decoded weights:

$$\mathcal{L}(\theta, W) = \sum_{i=1}^{M} \sum_{x \in P_i} \left( \text{SIREN}\left(x; \hat{W}_i\right) - \text{SIREN}\left(x; W\right) \right)^2, \tag{12}$$

$$\hat{W}_i = \text{DEC}_\theta(z_i), \quad z = \text{ENC}_\theta(W). \tag{13}$$

Our INR2ARRAY encoder architecture uses multiple weight-space self-attention layers followed by a weight-space cross-attention layer CA, which is especially well-suited for producing $M \times d$ permutation invariant arrays. For the decoder, we use the hypernetwork design of Sitzmann et al. [37], where the hypernetworks are conditioned on the spatial latent $z_i \in \mathbb{R}^d$. In principle, one can also design an invariant encoder using (non-attentive) NFN layers, and we will compare both options in our experiments.

## 5 Experiments

Our experiments broadly involve two types of weight-space datasets: (1) datasets of neural networks (e.g., CNN image classifiers) trained with varying hyperparameters, where the goal is to model interesting properties of the trained networks (e.g., generalization) from their weights; and (2) datasets containing the weights of INRs each representing a single data signal, such as an image. Tasks including editing INR weights to modify the represented signal, or predicting target information related to the signal (e.g., image class). For our INR experiments, we construct datasets for MNIST [25], FashionMNIST [42], and CIFAR-10 [23] following the procedure of Zhou et al. [46] exactly.

### 5.1 INR classification with INR2ARRAY

Classifying the signal represented by an INR directly from its weights is a challenging problem, with current approaches requiring that all INRs share initialization [10, 3, 6]. In the "vanilla" setting where INRs can be initialized and trained independently, state-of-the-art methods struggle to classify signals from even simple datasets [31, 46]. We train INR2ARRAY to learn a latent representation of INR weights that can be used for more effective classification in this vanilla setting.

We implement both NFT (weight-space self-attention and cross-attention) and $\text{NFN}_{\text{NP}}$ [46] variants of the INR2ARRAY encoder, and distinguish the two variants by superscripts: INR2ARRAY$^{\text{NFT}}$ and INR2ARRAY$^{\text{NFN}}$. For the decoder, we use the hypernetwork design of Sitzmann et al. [37], where the hypernetworks are conditioned on the spatial latent $z_i \in \mathbb{R}^d$. Appendix B.3 describes the implementation and training of each variant in full detail.

We train separate encoders for each INR dataset shown in Table 6 of the appendix, which reports the reconstruction mean square error (Eq 12) on test INR inputs. The NFT encoder consistently achieves lower reconstruction error than the NFN encoder, and INR2ARRAY$^{\text{NFT}}$ also produces qualitatively better reconstructions than INR2ARRAY$^{\text{NFN}}$ (Figure 4), confirming that NFTs enable higher quality encoding than previous neural functionals.

Once trained, the INR2ARRAY encoder maps INR weights to compact latent arrays that represent the INR's contents. Downstream tasks such as classifying the INR's signal (e.g., image) can now be performed directly on these $\mathcal{S}_{\text{NP}}$-invariant latent arrays $z \in \mathbb{R}^{M \times d}$. Concretely, in $K$-way

Table 1: Test accuracy (%) for weight-space INR classification in the MNIST, FashionMNIST, and CIFAR-10 datasets. INR2ARRAY significantly improves over current state-of-the-art results in this setting. For the NFN baseline [46] we report the higher performance out of their NP and HNP variants. Note that the MNIST and FashionMNIST results reported for DWS in Navon et al. [31] are on their own independently constructed INR datasets, while we use the datasets from Zhou et al. [46] for all methods for consistent comparison. Uncertainties indicate standard error over three runs.

|  | MNIST | FashionMNIST | CIFAR-10 |
|---|---|---|---|
| INR2VEC [6] | $19.1 \pm 0.18$ | $23.3 \pm 0.36$ | $16.7 \pm 0.24$ |
| DWS [31] | $74.4 \pm 0.14$ | $64.8 \pm 0.69$ | $41.5 \pm 0.43$ |
| NFN [46] | $92.9 \pm 0.38$ | $75.6 \pm 1.07$ | $46.6 \pm 0.13$ |
| Spatial NFN | $92.9 \pm 0.46$ | $70.8 \pm 0.53$ | $45.6 \pm 0.11$ |
| NFT | $89.9 \pm 1.03$ | $72.7 \pm 0.05$ | $44.8 \pm 0.32$ |
| INR2ARRAY$^{\text{NFN}}$ (Ours) | $94.6 \pm 0.00$ | $76.7 \pm 0.00$ | $45.4 \pm 0.00$ |
| INR2ARRAY$^{\text{NFT}}$ (Ours) | $\mathbf{98.5 \pm 0.00}$ | $\mathbf{79.3 \pm 0.00}$ | $\mathbf{63.4 \pm 0.00}$ |

classification the trained encoder $\text{ENC} : \mathcal{W} \to \mathbb{R}^{M \times d}$ can be composed with any classification head $f : \mathbb{R}^{M \times d} \to \mathbb{R}^K$ to form an $\mathcal{S}_{\text{NP}}$-invariant classifier $f \circ \text{ENC} : \mathcal{W} \to \mathbb{R}^K$. Since the encoder is already trained by the reconstruction objective, we only train $f$ during classification and keep the encoder fixed. We choose to implement $f$ using a Transformer classification head [41], which views the latent array input as a length-$M$ sequence of $d$-dimensional vectors.

Existing INR classifiers, which we refer to as DWS [31] and NFN [46], are permutation-invariant architectures that directly map input weights to predicted labels. We also create a modified NFN classifier (Spatial NFN) which produces an $M \times d$ array of activations before a convolutional classifier head, to test the impact of spatial latents independent of the INR2ARRAY training process. Appendix B.4 describes the setup for each method in further detail. Finally, we measure the performance of NFT classifiers without INR2ARRAY.

Table 1 shows that INR2ARRAY significantly improves classification test accuracies across the board. In addition, the best performance is achieved by implementing the encoder using our attention-based layers (NFT) compared to linear weight-space layers (NFN). Notably, INR2ARRAY$^{\text{NFT}}$ achieves a test accuracy of $64.4\%$ on CIFAR-10 (the hardest dataset), an improvement of $+17\%$ over the previous best result of $46.6\%$ by NFNs. They also achieve an MNIST test accuracy of $98.5\%$, up from $92.9\%$ by NFNs and $85.7\%$ by DWS. The vanilla NFT performance is comparable to DWS and NFNs (better than the former and worse than the latter), suggesting the importance of INR2ARRAY in addition to the architecture. Finally, the Spatial NFN fails to improve performance over standard NFN classifiers, implying that the particular objective of INR2ARRAY is crucial to make spatial latent spaces useful. The results show that learning a latent array representation of INR weights using INR2ARRAY and NFTs can greatly improve performance on downstream INR tasks like classification.

### 5.1.1 Term-by-term ablation

Here we investigate the importance of each term in our weight space self-attention layer (Eqs. 4-6). The first two terms of SA, illustrated by Figure 2, amount to self-attention between the rows and columns of *adjacent* weights, while the third term is a *global* self-attention between features from all layers. Note that in practice we use a tractable approximation of the third term described in Sec 3.1. For this ablation experiment we either keep only the first two terms (**AdjacentSA**) or only the third term (**GlobalSA**). **FullSA** denotes the original layer.

Table 2: Test classification accuracy of INR2ARRAY$^{\text{NFT}}$ when ablating the terms of SA (Eqs.4-6).

| Model | MNIST |
|---|---|
| FullSA | 98.5 |
| AdjacentSA | 97.6 |
| GlobalSA | 37.4 |

Table 2 shows the results of this ablation experiment on INR2ARRAY$^{\text{NFT}}$ performance in the MNIST classification task. We see that our full layer (FullSA) performs best, ablating the final term (AdjacentSA) only slightly degrades performance, and ablating the first two terms (GlobalSA) drastically harms performance. This is interesting since the first two terms are necessary to achieve *minimal equivariance*, while the third term alone is not minimally

Table 3: Test mean squared error to the target visual transformation for five INR editing tasks. NFTs generally achieve lower test error than NFN variants. Uncertainties indicate standard error over three seeds.

|  | NFN$_{\text{HNP}}$ [46] | NFN$_{\text{NP}}$ [46] | NFT (Ours) |
|---|---|---|---|
| MNIST (erode) | $0.0228 \pm 0.0003$ | $0.0223 \pm 0.0000$ | $\mathbf{0.0194 \pm 0.0002}$ |
| MNIST (dilate) | $0.0706 \pm 0.0005$ | $0.0693 \pm 0.0009$ | $\mathbf{0.0510 \pm 0.0004}$ |
| MNIST (gradient) | $0.0607 \pm 0.0013$ | $0.0566 \pm 0.0000$ | $\mathbf{0.0484 \pm 0.0007}$ |
| FashionMNIST (gradient) | $0.0878 \pm 0.0002$ | $0.0870 \pm 0.0001$ | $\mathbf{0.0800 \pm 0.0002}$ |
| CIFAR (contrast) | $0.0204 \pm 0.0000$ | $0.0203 \pm 0.0000$ | $0.0200 \pm 0.0002$ |

Table 4: (Small CNN Zoo [40] benchmark.) Rank correlation $\tau$ for predicting the generalization of CNN image classifiers with unseen hyperparameters trained CIFAR-10-GS and SVHN-GS (GS=grayscale). The NFT outperforms NFN$_{\text{NP}}$ and hand-picked features (STATNN), while NFN$_{\text{HNP}}$ performs best overall. Uncertainties indicate standard error over two runs.

|  | NFN$_{\text{HNP}}$ [46] | NFN$_{\text{NP}}$ [46] | STATNN [40] | NFT (Ours) |
|---|---|---|---|---|
| CIFAR-10-GS | $\mathbf{0.934 \pm 0.001}$ | $0.922 \pm 0.001$ | $0.915 \pm 0.002$ | $0.926 \pm 0.001$ |
| SVHN-GS | $\mathbf{0.931 \pm 0.005}$ | $0.856 \pm 0.001$ | $0.843 \pm 0.000$ | $0.858 \pm 0.000$ |

equivariant (but helps propagate information between any two weight matrices). The results emphasize the importance of designing our layer to achieve minimal equivariance rather than naively applying self-attention to the weights.

## 5.2 Editing INRs

We also evaluate NFTs on editing INR weights to alter their signal, e.g., to modify the represented image. The goal of this task is to edit the weights of a trained SIREN to alter its encoded image, expressed as a difference $W' \leftarrow W + \Delta(W)$. Neural functionals are trained to learn $\Delta(\cdot)$ that achieves some desired visual transformation, such as dilating the image. Permutation equivariance is a useful inductive bias here since if $\Delta(W)$ is the desired edit for $W$, then for any neuron permutation $\sigma$ the desired edit to $\sigma W$ is $\sigma \Delta(W)$. In addition to the MNIST dilation and CIFAR contrast tasks from Zhou et al. [46], we also introduce several new editing tasks: MNIST erosion, MNIST gradient, and FashionMNIST gradient. Gradient tasks roughly anount to edge detection; Figure 5 in the appendix shows sample inputs and targets for each editing task.

We compare NFT editing performance against the two NFN variants [46], with each method using the a similar number of parameters ($\sim$ 7M). For full training details, see Appendix B.2). Table 3 shows the test mean square error (MSE) between the edited INR and the ground truth visual transformation. NFTs consistently outperform NFNs on most INR editing tasks. In addition, Table 5 in the appendix shows that NFTs generally obtain both lower training and test error, indicating that the attention-based architecture enables greater expressivity. Figure 5 shows qualitative samples produced by each editing method on each task.

## 5.3 Predicting CNN classifier generalization

Whereas the previous experiments have focused on the weight spaces of INRs (which are implemented by small MLPs), we would also like to evaluate how NFTs process other weight spaces such as those belonging to convolutional neural network classifiers. Large-scale empirical studies have produced datasets of trained classifiers under different hyperparameter settings [40, 11], enabling a data-driven approach to modeling generalization from their weights, which could lead to new insights. Prior methods for predicting classifier generalization typically rely on extracting hand-designed features from the weights before using them to predict the test accuracy [18, 43, 40, 19, 28].

We now study how NFTs can model generalization from raw weights using the Small CNN Zoo [40], which contains thousands of CNNs trained on image classification datasets. In addition to comparing against the two neural functional variants NFN$_{\text{NP}}$ and NFN$_{\text{HNP}}$ [46], we also show the performance of

a hand-designed features approach called STATNN [40]. Each generalization predictor is trained on thousands of CNN weights produced under varying initialization and optimization hyperparameters, and is then tested on held out weights produced using unseen hyperparameters. Appendix B.1 describes the experimental setup in full detail.

Table 4 shows the test performance of each method using the Kendall rank correlation coefficient $\tau$ [20]. Across both datasets, neural functionals operating on raw weights outperform the hand-designed features baseline STATNN. NFTs slightly outperform $\text{NFN}_{\text{NP}}$ on each dataset, while $\text{NFN}_{\text{HNP}}$ achieves the best performance overall. Note that both $\text{NFN}_{\text{NP}}$ and NFTs use layers that assume input and output neurons are permutable (NP). Although we use input/output position encoding to remove the stronger NP assumptions, the results suggest that HNP designs may be naturally better suited to this task.

## 6 Related Work

It is well known that the weight spaces of neural networks contain numerous symmetries, i.e., transformations that preserve the network's behavior [17]. Permutation symmetries in particular have been studied in the context of neural network loss landscapes [14, 4, 12] and weight-space merging [38, 1]. However, most prior methods for processing weight space objects do not explicitly account for these symmetries [2, 26, 15, 24, 45, 7, 21], although some have tried to encourage permutation equivariance through data augmentation [33, 29]. Another workaround approach is to explicitly restrict the weight space being considered by, for example, fixing the initialization of all networks being processed [6] or meta-learning modulations of a set of base parameters [10, 3]. Instead, we study the problem of encoding permutation symmetries into the neural functional itself, without any restrictions on the weight space being processed.

There are a variety of methods that incorporate symmetries into deep neural network architectures [36, 22, 13]. Examples include (group) convolutions for images [25, 5] and permutation equivariant architectures for general set-structured inputs [34, 44, 16, 39, 27]. Most directly related to our work are that of Navon et al. [31] and Zhou et al. [46], who introduce linear layers equivariant to the neuron permutation symmetries of feedforward networks. We extend their approach by introducing nonlinear equivariant layers based on the attention mechanism, and use them to construct NFTs.

## 7 Conclusion

This work introduces neural functional transformers, a novel class of weight-space models designed with neuron permutation symmetries in mind. Our approach extends recent work on permutation-equivariant weight-space models using nonlinear self-attention and cross-attention layers, which can enable greater expressivity compared to existing linear layers.

We empirically evaluate the effectiveness of NFTs on weight-space tasks involving datasets of trained CNN classifiers and implicit neural representations (INRs). Operating on weights alone, NFTs can predict the test accuracy of CNN classifiers, modify the content of INRs, and classify INR signals. We also use NFTs to develop INR2ARRAY, a method for mapping INR weights to compact and $\mathcal{S}_{\text{NP}}$-invariant latent representations, which significantly improve performance in downstream tasks such as INR classification.

Some limitations of NFTs include increased computational costs from self-attention relative to linear weight-space layers, and the difficulty of training large NFT architectures stably. Future work may address these limitations, and explore additional applications of NFTs such as for learned optimization or weight-space generative modeling. Overall, our work highlights the potential of attention-based weight-space layers offers a promising direction for the development of more expressive and powerful neural functional architectures.

## Acknowledgments and Disclosure of Funding

We thank Adriano Cardace and Yoonho Lee for interesting discussions and suggestions related to the development of this paper. AZ and KB are supported by the NSF Graduate Research Fellowship Program. This project was supported by Juniper Networks and ONR grant N00014-22-1-2621.

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

# A Weight-space self-attention

## A.1 Proof of minimal equivariance

We repeat the claim about minimal equivariance for weight-space self-attention and layer position encodings SA ∘ LAYERENC : $\mathcal{W}^c \to \mathcal{W}^c$:

**Theorem 3.1** (Minimal equivariance)**.** *The combined layer* SA ∘ LAYERENC *is **minimally $\mathcal{S}_{NP}$-equivariant**.*

*Proof.* We first show $\mathcal{S}_{NP}$-equivariance, before checking that it minimally equivariant.

**Equivariance**. It is straightforward to check that LAYERENC is equivariant to any permutation that preserves the layer index (which neuron permutations do). At a high level, we check that SA is $\mathcal{S}_{NP}$-equivariant by verifying that $\text{SA}(\sigma W)^{(i)}_{jk} = \text{SA}(W)^{(i)}_{\sigma_i^{-1}(j), \sigma_{i-1}^{-1}(k)}$. We proceed term by term.

First, it will be useful to show that ATTN (Eq. 3) is equivariant to permutations $\pi \in S_d$ of each key, query, and value input. In particular, let $[\pi q][i] = q[\pi^{-1}(i)]$ permute the entries of the input vectors:

$$\text{ATTN}\left(\pi q, \{\pi k_p, \pi v_p\}_{p=1}^N\right)[i] = \left[\sum_p \left(\frac{\exp(q \cdot k_p)}{\sum_{p'} \exp(q \cdot k_{p'})}\right)(\pi v_p)\right][i] \quad (14)$$

$$= \sum_p \left(\frac{\exp(q \cdot k_p)}{\sum_{p'} \exp(q \cdot k_{p'})}\right) v_p[\pi^{-1}(i)] \quad (15)$$

$$= \text{ATTN}(q, \{k_p, v_p\}_{p=1}^N)[\pi^{-1}(i)] \quad (16)$$

Note that in the first equality we use $\pi q \cdot \pi k = q \cdot k$.

Since the key, query, and value projections are only linear projections along the channel dimension, $K(\sigma W) = \sigma K(W)$, where $K$ denotes all keys produced by linearly projection of $W$ by $\theta_K$ (and similarly for $Q, V$). Then consider, for example, how the first term (Eq 4) changes under $W \mapsto \sigma W$:

$$\text{Attn}\left(\sigma Q^{(i)}_{j,:}, \left\{(\sigma K, \sigma V)^{(i-1)}_{:,q}\right\}_q \bigcup \left\{(\sigma K, \sigma V)^{(i)}_{p,:}\right\}_p\right)[k] = \quad (17)$$

$$\text{Attn}\left(\sigma_{i-1} Q_{\sigma_i^{-1}(j),:}, \left\{(\sigma_{i-1} K^{(i-1)}_{:,q}, \sigma_{i-1} V^{(i-1)}_{:,q})\right\}_q \bigcup \left\{(\sigma_{i-1} K^{(i)}_{p,:}, \sigma_{i-1} V^{(i)}_{p,:})\right\}_p\right)[k] \quad (18)$$

Note the distinction between neuron permutations of the weight space $[\sigma Q]$ and permutations of individual vectors like $\sigma_{i-1} Q_{p,:}$, and that permutations along the KV set indices (e.g., $q$ in the first set) are ignored since set order is irrelevant. Using the equivariance of ATTN (Eqs. 14-16):

$$\text{ATTN}\left(Q^{(i)}_{\sigma_i^{-1}(j),:}, \left\{(K, V)^{(i-1)}_{:,q}\right\}_q \bigcup \left\{(K, V)^{(i)}_{p,:}\right\}_p\right)[\sigma_{i-1}^{-1}(k)] \quad (19)$$

Notice that compared to Eq. 4 the output indices have changed $j \mapsto \sigma_i^{-1}(j)$ and $k \mapsto \sigma_{i-1}^{-1}(k)$, showing that the first term is equivariant. We can repeat this argument in a similar fashion for the second term (Eq. 5), while the third term (Eq. 6) applies ATTN by treating each weight as an individual token, so its equivariance follows immediately from the permutation equivariance of ATTN. This verifies that all terms of SA are $\mathcal{S}_{NP}$-equivariant.

**Minimal equivariance.** Here our goal is to show *non*-equivariance to permutations $\tau \in S_{\dim(\mathcal{W})}$ but $\tau \notin \mathcal{S}_{NP}$ (false symmetries). That is, we should find some input $W$ such that $\text{SA}(\text{LAYERENC}(\tau W)) \neq \tau \text{SA}(\text{LAYERENC}(W))$, for some parameters $\{\theta_Q, \theta_K, \theta_V\}$ and $\{\phi^{(i)}\}$.

It will be useful to define the actions of permutations in $S_{\dim(\mathcal{W})}$ on the indices of the weights, rather than the weights themselves. The index-space for a weight space $\mathcal{W}$ is defined as a set of 3-tuples representing layer number, row, and column, respectively:

$$\mathbb{I} = \{(i, j, k) \mid i = [\![1..L]\!], j = [\![1..n_i]\!]; k = [\![1..n_{i-1}]\!]\}. \quad (20)$$

For $\alpha \in \mathbb{I}$, we use the subscripts $\ell, r, c$ to denote the layer, row, and column indices respectively. That is, $\alpha = (\alpha_\ell, \alpha_r, \alpha_c) = (i, j, k)$.

Notice that the set of arbitrary permutations $S_{\dim(\mathcal{W})}$ is simply the set of all bijections $\tau : \mathbb{I} \to \mathbb{I}$ (any index can be arbitrarily permuted to any other index), while $\mathcal{S}_{\text{NP}}$ is the subgroup of bijections $\sigma$ that can be written as:

$$\sigma(i, j, k) = (i, \sigma_i(j), \sigma_{i-1}(k)), \quad \forall (i, j, k) \in \mathbb{I}. \tag{21}$$

False symmetries can broadly be categorized into three groups. In fact, we can verify that any permutation $\tau$ that fails to satisfy one of these three cases must in $\mathcal{S}_{\text{NP}}$:

1. **False symmetry that permutes weights across layers.** There exists an $(i, j, k) \in \mathbb{I}$ such that $\tau(i, j, k)_\ell \neq i$.
2. **False symmetry where row and column permutations are not independent.** Suppose a false symmetry $\tau$ does not fall under case (1), i.e., it preserves the layer index. Then it may still fail to have independent permutations of row and columns. In this case, either there is an $(i, j, k)$ and $q$ such that $\tau(i, j, k) = (i, j', k')$, but $\tau(i, j, q)_r \neq j'$. Or there is an $(i, j, k)$ and $p$ such that $\tau(i, j, k) = (i, j', k')$ but $\tau(i, p, k)_c \neq k'$.
3. **Adjacent permutations decoupled.** If a false symmetry $\tau$ is not one of the two cases above, there must be an $(i, j, k) \in \mathbb{I}$ and $q$ such that $\tau(i, j, k) = (i, j', k')$ but $\tau(i - 1, k, q)_r \neq k'$. That is, an instance where one row pair of layer $i - 1$ do not permute the same way as the columns of layer $i$.

We can now check non-equivariance for $\tau$ in each case. To simplify the notation here we will assume $c = 1$ although the general case is similar:

1. If $\tau$ does not preserve layer index, LAYERENC will be non-equivariant since each layer has a different encoding.
2. Suppose there is an $(i, j, k)$ and $q$ such that $\tau(i, j, k) = (i, j', k')$, but $\tau(i, j, q)_r \neq j'$, and let $\theta_Q = \theta_K = \theta_V = I$. Now consider the input $W \in \mathcal{W}$ such that all weights are 0 except for $W_{jk}^{(i)} = W_{jq}^{(i)} = 1$. Then we can check that $[\tau\text{SA}(W)]_{i,j',k'} \neq \text{SA}(\tau W)_{i,j',k'}$, so the layer is not equivariant to $\tau$. We can construct a similar argument for the other case, where there is an $(i, j, k)$ and $p$ such that $\tau(i, j, k) = (i, j', k')$ but $\tau(i, p, k)_c \neq k'$.
3. In this case there must be an $(i, j, k) \in \mathbb{I}$ and $q$ such that $\tau(i, j, k) = (i, j', k')$ but $\tau(i - 1, k, q)_r \neq k'$. Again let $\theta_Q = \theta_K = \theta_V = I$ and consider $W \in \mathcal{W}$ such that all weights are 0 except for $W_{jk}^{(i)} = W_{kq}^{(i-1)} = 1$. Then, similar to the above case, we can verify that $[\tau\text{SA}(W)]_{i,j',k'} \neq \text{SA}(\tau W)_{i,j',k'}$, so the layer is not equivariant to such $\tau$.

Taken together, the three cases above show that although SA $\circ$ LAYERENC is $\mathcal{S}_{\text{NP}}$-equivariant, it is not equivariant to any other permutations $\tau \notin \mathcal{S}_{\text{NP}}$. Hence the layer is *minimally* equivariant. $\quad\square$

### A.2 Full description including biases

Let $\mathcal{B}$ be the space of biases, and $\mathcal{U} = \mathcal{W} \times \mathcal{B}$ the full space of both weights and biases. Then the full weight-space self-attention layer SA $: \mathcal{U}^c \to \mathcal{U}^c$ is defined, for input $U = (W, b)$:

$$\text{SA}(U; \theta_Q, \theta_K, \theta_V) = (Y(U), z(U)), \tag{22}$$

where $Y(U) \in \mathcal{W}^c$ and $z(U) \in \mathcal{B}^c$, with:

$$Y(W,b)^{(i)}_{jk} = \text{ATTN}\left(Q^{(i)}_{j,:}, KV_1\right)_k + \text{ATTN}\left(Q^{(i)}_{:,k}, KV_2\right)_j + \text{ATTN}\left(Q^{(i)}_{jk}, KV_3\right), \quad (23)$$

$$z(W,b)^{(i)}_j = \text{ATTN}\left(q^{(i)}, KV_2\right)_j + \text{ATTN}\left(q^{(i)}_j, KV_3\right), \text{ where} \quad (24)$$

$$KV_1 = \left\{ (K,V)^{(i-1)}_{:,q} \right\}_{q=1}^{n_{i-2}} \bigcup \left\{ (K,V)^{(i)}_{p,:} \right\}_{p=1}^{n_i} \bigcup \left\{ (k,v)^{(i-1)} \right\} \quad (25)$$

$$KV_2 = \left\{ (K,V)^{(i)}_{:,q} \right\}_{q=1}^{n_{i-1}} \bigcup \left\{ (K,V)^{(i+1)}_{p,:} \right\}_{p=1}^{n_{i+1}} \bigcup \left\{ (k,v)^{(i)} \right\} \quad (26)$$

$$KV_3 = \left\{ (K,V)^{(s)}_{pq} \,\Big|\, \forall s,p,q \right\} \bigcup \left\{ (k,v)^{(s)}_p \,\Big|\, \forall s,p \right\}, \text{ and} \quad (27)$$

$$Q^{(i)}_{j,k} := \theta_Q W^{(i)}_{jk}, \quad K^{(i)}_{j,k} := \theta_K W^{(i)}_{jk}, \quad V^{(i)}_{j,k} := \theta_V W^{(i)}_{jk}, \quad (28)$$

$$q^{(i)}_j := \theta_Q b^{(i)}_j, \quad k^{(i)}_j := \theta_K b^{(i)}_j, \quad v^{(i)}_j := \theta_V b^{(i)}_j. \quad (29)$$

We use $KV_1, KV_2, KV_3$ to denote the three sets of key-value pairs involved in the layer computation and distinguish the linear projections of weights and biases using uppercase $(Q, K, V)$ and lowercase $(q, k, v)$, respectively.

## B  Experiment details

Recall from Section 3.4 that NFTs consist of "blocks" that include weight-space self attention, layer normalization, and feedforward MLPs. Additionally, the first layer is typically a projection from the input channel dimensionality to the hidden channel dimensionality, for which we use random Fourier features.

| Hyperparameters | Predicting gen. | Editing INRs | INR2ARRAY |
|---|---|---|---|
| # blocks | 4 | 4 | 6 |
| # channels | 256 | 256 | 256 |
| MLP hidden dim | 1024 | 512 | 1024 |
| Fourier scale | 3 | 3 | 3 |
| Fourier size | 128 | 128 | 128 |
| # attn heads | 4 | 8 | 4 |
| dropout p | 0.1 | 0.1 | 0 |
| invariant layer | HNP | – | CA |
| CA $M$ | – | – | 16 |
| CA dim | – | – | 256 |
| Optimizer | Adam | Adam | AdamW |
| Learning rate | 0.001 | 0.001 | 0.0001 |
| Weight decay coeff | 0 | 0 | 0.01 |
| LR warmup steps | 10K | 10K | 10K |
| Total params | 46M | 7M | 22M |
| Training steps | 75K | 50K | 200K |
| Training time | 5H | 2H | 13H |

Here we summarize the hyperparameters (and other training information) for the NFTs in each experiment in this paper. NFTs are described by the following hyperparameters:

- # blocks: Number of equivariant weight-space blocks $\text{BLOCK}(\cdot)$
- # channels: channel dimension $c$ used by the blocks.
- MLP hidden dim: the hidden layer size of the feedforward MLPs within each block
- Fourier scale: standard deviation of random vectors used to compute random Fourier features
- Fourier size: number of random fourier features
- # attn heads: Number of attention heads used by weight-space self-attention.
- dropout p: Dropout probability for attention matrix and in feedforward networks.
- invariant layer (for $\mathcal{S}_{\text{NP}}$-invariant NFTs): Whether the invariant pooling is done with summation (HNP/NP [46]) or cross-attention (ours).
- CA $M$: If using cross attention, the number of learned query embeddings $e_1, \cdots, e_M$.
- CA dim: Dimension of vectors going into cross attention.

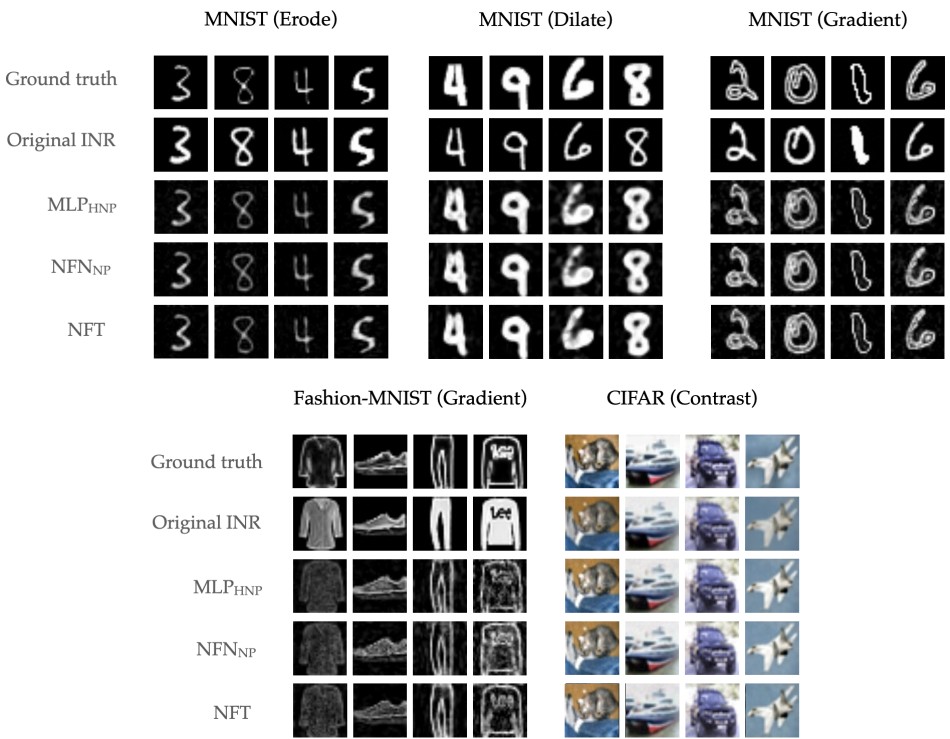

Figure 5: Samples of target image transformations, as well as original and edited INRs for each editing task in Table 3. Despite the fact that the NFT quantitatively outperforms the NFN$_{HNP}$ and NFN$_{NP}$, compelling qualitative differences in the edited images can be difficult to identify due to small image size.

## B.1 Generalization prediction

Each Small CNN Zoo dataset that we consider (CIFAR-10-GS and SVHN-GS) contains the weights of 30000 CNNs trained with varying hyperparameters. The weights are split into 15000 for training and validation and the rest for testing. We use 20% of the non-test data for validation, and train on the remaining 12000. We use the validation data to sweep over Fourier scales of 3 or 10 and try both HNP (summation) or CA (cross attention) layers to achieve invariance.

## B.2 Style editing

We implement four visual transformations: erode reduces the size of objects in an image by removing pixels on the boundary; dilate adds pixels to object boundaries and can be thought of as the opposite of erode; morphological gradient performs edge detection by computing the difference between dilation and erosion; and contrast magnifies color differences in an image, making objects more distinguishable. Figure 5 shows visual samples of the input and target for each task.

## B.3 INR2ARRAY

We split the SIRENs of each dataset into training, validation, and testing sets, and train the encoder and decoder using only the training set. We use the validation error to sweep over the following hyperparameters (for both the NFT and NFN variants): # blocks (4 vs 6), # channels (256 vs 512), MLP hidden dim (512 vs 1024), Fourier scale (3 vs 10), # attn heads (4 vs 8), and dropout (0.1 vs 0). After sweeping these hyperparameters, we use the best hyperparameter configuration to train the encoder and decoder and do early stopping with the validation error. The encoder portion of the best checkpoint can then be used to produce latent array representations of INR inputs.

Table 5: Mean squared error to ground truth visual transformation for five INR editing tasks. NFTs generally achieve lower both lower training and test error than NFN variants without using more parameters, suggesting greater expressivity from attention. Uncertainties indicate standard error over three seeds.

|  |  | $\text{NFN}_{\text{HNP}}$ [46] | $\text{NFN}_{\text{NP}}$ [46] | NFT (Ours) |
|---|---|---|---|---|
| MNIST (erode) | Train | $0.0217 \pm 0.0004$ | $0.0212 \pm 0.0002$ | $0.0195 \pm 0.0004$ |
|  | Test | $0.0225 \pm 0.0001$ | $0.0223 \pm 0.0000$ | $\mathbf{0.0194 \pm 0.0002}$ |
| MNIST (dilate) | Train | $0.0575 \pm 0.0010$ | $0.0671 \pm 0.0017$ | $0.0473 \pm 0.0006$ |
|  | Test | $0.0656 \pm 0.0000$ | $0.0693 \pm 0.0009$ | $\mathbf{0.0510 \pm 0.0004}$ |
| MNIST (gradient) | Train | $0.0536 \pm 0.0020$ | $0.0547 \pm 0.0009$ | $0.0474 \pm 0.0005$ |
|  | Test | $0.0552 \pm 0.0003$ | $0.0566 \pm 0.0000$ | $\mathbf{0.0484 \pm 0.0007}$ |
| FashionMNIST (gradient) | Train | $0.0851 \pm 0.0012$ | $0.0853 \pm 0.0008$ | $0.0795 \pm 0.0009$ |
|  | Test | $0.0859 \pm 0.0001$ | $0.0870 \pm 0.0001$ | $\mathbf{0.0800 \pm 0.0002}$ |
| CIFAR (contrast) | Train | $0.0191 \pm 0.0002$ | $0.0185 \pm 0.0002$ | $0.0192 \pm 0.0005$ |
|  | Test | $0.0204 \pm 0.0000$ | $0.0203 \pm 0.0000$ | $0.0200 \pm 0.0002$ |

Table 6: Mean squared error (MSE) of INR2ARRAY reconstructions on test INR weights. Using an NFT encoder in INR2ARRAY achieves better reconstruction error, produces higher quality samples, and leads to better downstream performance.

|  | INR2ARRAY$^{\text{NFN}}$ | INR2ARRAY$^{\text{NFT}}$ |
|---|---|---|
| MNIST | $0.046 \pm 0.001$ | $\mathbf{0.027 \pm 0.003}$ |
| FashionMNIST | $0.085 \pm 0.006$ | $\mathbf{0.070 \pm 0.006}$ |
| CIFAR-10 | $0.117 \pm 0.008$ | $\mathbf{0.036 \pm 0.006}$ |

## B.4 INR Classification

After training INR2ARRAY on a given dataset, we select the checkpoint with the smallest reconstruction error on the validation set, and use the encoder to produce a $16 \times 256$ latent array representation for any input SIREN. We view the latent as a length-16 sequence of 256-dim vectors, and project each vector into $\mathbb{R}^{512}$ before feeding the array into Transformer encoder with 12 blocks. This produces an output $z^o \in \mathbb{R}^{16 \times 512}$; we take the first output $z_0^o \in \mathbb{R}^{512}$ and feed it into a 2-layer MLP with 512 hidden units that produces the classification logits. We train with cross-entropy loss for 100 epochs, using the AdamW optimizer and mixup augmentation. Using the validation accuracy, we sweep over weight decay (0.1 vs 0.01) and mixup alpha (0 vs 0.2 vs 1) and perform early stopping.

Table 7: Classification train and test accuracies (%) on datasets of image INRs (MNIST, FashionMNIST, and CIFAR-10). INR2ARRAY outperforms prior state-of-the-art results in this setting (DWS and NFN). For the NFN baseline [46] we report the higher performance out of their NP and HNP variants. Uncertainties indicate standard error over three runs.

|  | MNIST | | FashionMNIST | | CIFAR-10 | |
|---|---|---|---|---|---|---|
|  | Train | Test | Train | Test | Train | Test |
| DWS [31] | – | $85.7 \pm 0.57$ | – | $65.5 \pm 0.48$ | – | – |
| NFN [46] | $96.2 \pm 0.24$ | $92.9 \pm 0.38$ | $84.1 \pm 1.02$ | $75.6 \pm 1.07$ | $61.0 \pm 6.60$ | $46.6 \pm 0.13$ |
| Spatial NFN (Ours) | $95.3 \pm 0.59$ | $92.9 \pm 0.46$ | $76.4 \pm 0.87$ | $70.8 \pm 0.53$ | $52.3 \pm 0.30$ | $45.6 \pm 0.11$ |
| INR2ARRAY$^{\text{NFN}}$ (Ours) | $100 \pm 0.00$ | $94.6 \pm 0.00$ | $92.8 \pm 0.00$ | $76.7 \pm 0.00$ | $98.5 \pm 0.00$ | $45.4 \pm 0.00$ |
| INR2ARRAY$^{\text{NFT}}$ (Ours) | $100 \pm 0.00$ | $\mathbf{98.5 \pm 0.00}$ | $97.6 \pm 0.00$ | $\mathbf{79.3 \pm 0.00}$ | $100 \pm 0.00$ | $\mathbf{63.4 \pm 0.00}$ |

