# OpenReview forum: "Neural Functional Transformers"
_NeurIPS.cc/2023/Conference — NeurIPS 2023 poster_

### Official Review · Reviewer_73nt · 2023-06-30

**Soundness:** 2 fair
**Presentation:** 3 good
**Contribution:** 2 fair
**Rating:** 6
**Confidence:** 4

**Summary:**

Designing architectures for processing the weights of neural networks is an exciting and relatively new area with great implications. This paper proposes a novel, attention-based architecture for processing NN weights. The proposed architecture is equivariant to the permutation symmetries of neural weight spaces. Furthermore, the authors propose INR2Array, a method for learning an invariant latent space for INR weights.


**Strengths:**

1. The paper is generally well-structured and well-written.
1. The authors propose a novel architecture for processing the weights of neural networks. The new attention-based architecture achieves comparable or superior results compared to baseline methods.
1. This is a natural and necessary extension of previous work on linear equivariant weight space layers.


**Weaknesses:**

My main concerns are the limited novelty w.r.t prior works and the missing/unfair evaluation results, which makes it difficult to assess the performance of the proposed approach w.r.t to prior work, and the effect of the different part of the proposed method (e.g., INR2Array^NFT vs. NFT).

1. While attention-based equivariant layers are a natural extension, the technical novelty of the paper is somewhat limited with respect to prior works ([1], [2], [3]).
1. Missing results for NFT without INR2Array for the INR classification task in Table 1. This is a necessary comparison for evaluating the effect of the main contributions of the paper, attention-based base layers for weight spaces, and INR2Array.
1. The evaluation against DWS in Table 1 is flawed and unfair: The results for DWS are taken from [1], which uses a single INR copy for each image. The results for other baselines and the proposed method use 10 INR copies. As discussed in Section 7 of [1], additional INR copies can significantly improve the performance. Also, a more direct evaluation should use the same INR dataset, as choices of the INR arch., hyperparameters, and training procedure can affect the results of methods trained using this INR dataset. Using different INR datasets can potentially greatly vary the results. Finally, the augmentation scheme used in this paper is different than the one used in [1].
1. Line 120: Need to provide an explanation/intuition for the provided claim.
1. The equivariance proof provided for Theorem 3.1 is insufficient. While I believe the results are valid, the claim (“To show this, we can expand the left hand side term-by-term using the definition of SA.”) should be extended, and more details are needed. Also, it would be beneficial to provide a concrete example (using a specific layer) to show equivariance.
1. As stated by the authors, the proposed layers are computationally expensive. Please provide runtime (or complexity) comparisons w.r.t linear equivariant weight-space layers.
1. Additional comments on experiments:
    - Please report the number of parameters for each method (In Table 1).
    - What is the NFN (and INR2Array^NFN) variant used in Table 1 (NP or HNP)?
    - It would be beneficial to include ablation on design choices, specifically the use of Random Fourier features and the use of Layer norm.
    - Please provide more details on the use of mixup augmentations: are you performing the mixup in weight space?
    - Also, it would be beneficial to compare the results of NFT with/without mixup augmentation.

Minor:
1. In Sec. 2, it would be clearer if the authors explicitly state that elements from the NP group, generally do not preserve the function defined by the set of weights.

[1] Navon et al., Equivariant architectures for learning in deep weight spaces, ICML 2023.

[2] Zhou et al., Permutation Equivariant Neural Functionals, 2023.

[3] Luigi et al., Deep Learning on Implicit Neural Representations of Shapes, ICLR 2023.



**Questions:**

1. Can the proposed Arch. be easily extended to the HNP design and setup? From Table 3 it appears this could be very beneficial for some tasks.
1. The authors stated the difficulty of training large NFT architectures stably as a limitation of the method (Section 7). Since learning over weight spaces is a relatively novel task, it would be very beneficial if the authors could elaborate on any limitations/findings/failure cases from their experiments in the Appendix.


**Limitations:**

Addressed, although more details would be beneficial, see Questions.

---

> ### Author Rebuttal · Authors · 2023-08-10
>
> Thank you for your careful review of our work. We will try to address your questions and feedback individually here:
>
> > the technical novelty of the paper is somewhat limited with respect to prior works ([1], [2], [3])
>
> In addition to extending the equivariant framework of [1,2], we highlight our theoretical analysis of minimal equivariance in our self attention layer (Theorem 3.1), which is notable relative to [1,2] since those focus on theoretical properties of linear layers, while our layer is nonlinear. Moreover, in Inr2Array we add a spatial latent structure which is crucial to achieve good performance on the 2D image INR tasks, so our contribution in Inr2Array is not merely combining equivariant layers with Inr2Vec.
>
> > Missing results for NFT without INR2Array for the INR classification task in Table 1.
>
> We will run this and report the results when they are available.
>
> > The results for DWS are taken from [1] [...] using different INR datasets can potentially greatly vary the results.
>
> We've put the corrected comparison in the top-level reply. The relative ordering of methods by performance remains broadly unchanged. We will also add the corrected results to the paper.
>
> > Line 120: Need to provide an explanation/intuition for the provided claim [“Self attention between these n_i + n_{i-2} vectors is only equivariant if if the rows and columns of W^{(i-1)} and W^{(i)} permute simultaneously”]
>
> We will add the following explanation:
> > Permuting the rows of $W^{(i-1)}$ and columns of $W^{(i)}$ corresponds to permuting the feature dimension of the inputs to Attn (Eq 3). If the features are permuted simultaneously, the pairwise dot products between any two vectors are preserved and the attention matrix is unchanged, producing an output whose features are simply permuted (equivariance). However, independent permutations of the Attn inputs will in general change the dot products and hence the attention matrix, breaking equivariance. For a more detailed explanation, see the proof of non-equivariance to false symmetries in Appendix A.1, in particular cases 2 and 3.
>
> The new diagram of SA that we have provided should also aid in illustrating this argument visually.
>
> > The claim [equivariance of SA] should be extended, and more details are needed
>
> See the expanded proof of this claim in the top-level reply, which we will add to the Appendix.
>
> > runtime (or complexity) comparisons w.r.t linear equivariant weight-space layers.
>
> Using the same (A40) GPU, INR2Array with NFT takes 2.79x times as long per training step compared to with NFNs (NP). We will add full runtime comparisons to the paper.
>
> > Please report the number of parameters for each method (In Table 1).
>
> We will add the number of parameters for each baseline method in Table 1. The number of parameters for the NFT method in each experiment is already reported in Appendix B. For quick reference:
>
> | Method            | # params                                    |
> |-------------------|---------------------------------------------|
> | DWS (32 chan)     | 0.5M (F/MNIST), 1M (CIFAR)                                      |
> | DWS (512 chan)    | 71M (F/MNIST), 134M (CIFAR)                 |
> | NFN_NP            | 45M                                         |
> | INR2Array (NFN)   | 31M                                         |
> | INR2Array (NFT)   | 22M                                         |
>
>
> > What is the NFN (and INR2Array^NFN) variant used in Table 1 (NP or HNP)?
>
> That used NP. We are running an HNP variant for comparison and will include its results in the revised paper
>
> > [Various questions about mixup and its effect]
>
> Mixup (with $\alpha=1$) is applied to the latent embeddings of INRs produced by Inr2Array–not the raw weights--for downstream INR classification.
>
> As suggested, we also tried removing mixup. The results suggest that mixup provides a very small benefit in most cases, and a modest benefit on CIFAR-10:
>
> | Dataset      | INR2Array^NFT | Without mixup |
> |--------------|---------------|---------------|
> | **MNIST**       | 98.5          | 98.3          |
> | **FashionMNIST** | 79.3          | 78.8          |
> | **CIFAR-10**     | 63.4          | 60.2          |
>
> We will include these details in the revised paper.
>
> > It would be clearer if the authors explicitly state that elements from the NP group,  generally do not preserve the function defined by the set of weights.
>
> We will modify Section 2 to make this point about Neuron Permutations clear.
>
> > [can] the proposed Arch. be easily extended to the HNP design?
>
> Yes, it is likely that an HNP extension of the NFT layers would be helpful in some settings, and should be feasible. Though it may perhaps be complicated to implement and computationally expensive in some settings.
>
> > It would be beneficial to include ablation on design choices [...] Authors could elaborate on any limitations/findings/failure cases
>
> Although we have not had the bandwidth to run these ablations yet, we will aim to do so and include the results in the revised paper, in addition to our findings on how to train large NFT architectures stably.

---

> > ### Comment · Reviewer_73nt · 2023-08-15
> > **Replying to Rebuttal**
> >
> > Thank you for your response and for providing additional results. I have some following questions and concerns:
> >
> > > Regarding the NFT without INR2Array results (reported in response to Reviewer 5PwT):
> >
> > It seems like the proposed architecture (NFT) achieves lower results compared to NFN on two out of the three tasks presented in the paper, namely "INR classification" and "predicting CNN classifier generalization". This result casts doubt on the effectiveness of the proposed architecture compared to previous methods. This is amplified since the proposed method is computationally expensive compared to, e.g., NFN as stated by the authors: "INR2Array with NFT takes 2.79x times as long per training step compared to with NFNs (NP)"
> >
> > > Regarding the results for DWS:
> >
> > The original DWS paper reported 85.7 for MNIST using a single INR per image. Could you please explain your results of 74.7 for a dataset with 10 INRs per image?

---

> > > ### Author Response · Authors · 2023-08-16
> > > **Thank you for input and additional questions!**
> > >
> > > >  lower results compared to NFN on two out of the three tasks
> > >
> > > Good point, this gets to the primary motivation/contribution of the NFT architecture. Our primary intuition motivating NFTs is that attention based layers can be more **expressive** than linear ones. Relative to DWS/NFN, we don't expect NFTs to possess any _additional_ inductive biases that would lead to better generalization on weight space tasks. So we would expect better training performance, but not necessarily better generalization (=difference between test and train performance). However, simply having powerful models that can fit the training data better can be advantageous, for example if one has a lot of data or the task is very challenging.
> > >
> > > On direct MNIST classification, NFTs achieve better training accuracy than NFNs with fewer parameters, while NFNs have better test performance. This result is congruent with what we expected above.
> > >
> > > With a richer objective and more challenging reconstruction task (Inr2Array), the advantage of the attention-based layers in NFTs becomes clearer. NFTs achieve a lower **training** reconstruction error than NFNs in Inr2Array, while using fewer parameters:
> > >
> > > | Training recon error | INR2Array (NFN) | INR2Array (NFT) |
> > > |---------------------------------|-----------------|-----------------|
> > > | MNIST                           | .035            | .016            |
> > > | FashionMNIST                    | .068            | .053            |
> > > | CIFAR-10                        | .093            | .026            |
> > >
> > > Unlike in classification, NFNs here evidently struggle to even fit the data in some cases (>3x higher training error on CIFAR-10), and qualitative reconstruction samples even on _training_ inputs confirm this visually. So NFTs lead to much better downstream task performance, especially on CIFAR-10. We will add these details and clarify the motivation in the paper.
> > >
> > > > The original DWS paper reported 85.7 for MNIST
> > >
> > > The MNIST INR datasets in [1] and [2] were constructed independently--"DWS=85.7" is reported for the dataset from [1], while the results in our work use the dataset from [2]. We used the INR datasets from [2] since they included CIFAR-10, the most challenging dataset in Table 1.
> > >
> > > Looking at the details of each paper, some differences include learning rates (5e-4 in [1] vs 5e-5 in [2]) and the use of early stopping in [1]. Optimization differences would lead to a different distribution of resulting INR weights. Adding multiple INRs per image has never hurt performance in any method on any dataset we've tried (as also confirmed by experiments in [1]), so the difference is likely due to the aforementioned differences in INR production.
> > >
> > > Our reported DWS numbers were obtained using the official DWS code for MNIST classification, but with the datasets swapped out. We also computed statistics for normalizing the inputs using the DWS code. For each architecture size we tried, we also swept learning rates according to the process reported in in [1] itself. We will continue to investigate what might be causing this discrepancy.
> > >
> > > [1] Navon et al., Equivariant architectures for learning in deep weight spaces, ICML 2023.
> > >
> > > [2] Zhou et al., Permutation Equivariant Neural Functionals, 2023.

---

> > > > ### Comment · Reviewer_73nt · 2023-08-17
> > > >
> > > > Thank you for providing more details and addressing most of my concerns.
> > > >
> > > > My main concern at the moment is that the paper does not reflect the primary motivation for NFTs very well. If you expect attention-based layers to outperform linear layers on more challenging tasks, then providing more empirical evidence for that is essential. At the moment, the paper presents fairly simple benchmarks already proposed in previous works (i.e., in the NFN and DWS papers), which mostly do not benefit from attention-base layers, both in terms of task performance (e.g., accuracy) and computational efficiency.

---

### Official Review · Reviewer_5PwT · 2023-07-05

**Soundness:** 2 fair
**Presentation:** 2 fair
**Contribution:** 2 fair
**Rating:** 6
**Confidence:** 3

**Summary:**

The paper studies an interesting problem of encoding weights of neural network using another network (Neural Functional Transformer or NFT). NFTs are built based on previous works, mainly DWS [31] and NFN [46], proposing permutation equivariant linear models to encode neural network weights. NFT's main contribution compared to these is a more expressive model based on Transformers that is permutation equivariant only to those weight permutations that result in the same function. While DWS already proposed such a minimal permutation equivariance, this paper achieves better performance due to leveraging Transformers and other tricks such as spatial INRs (INR2ARRAY).
Overall, NFTs achieve better performance when encoding the weights of INRs as well as competitive performance in other tasks.

**Strengths:**

1. The topic of encoding neural network weights is very interesting with a lot of potential.
2. The idea of combining Transformers with minimal equivariance seems as a powerful and promising way to approach this problem.
3. The improvements in most experiments are significant.
4. The paper is written and presented relatively well.


**Weaknesses:**

While I really like the problem the paper tackles and using Transformers with minimal equivariance, unfortunately there are many weak points described below. I will be willing to raise my score if these are addressed.

1. Intuitive explanation of the difference between Eq. 4 and 5 is missing (why the union of sets is used and how is it implemented, by concatenation?) Perhaps, some illustration would help. Moreover, it would be very useful to see ablation results with each of the terms (4,5,6) ablated in the model.

2. The paper mentioned that the weight space can include different features such as weights and gradients, but the details of what features are actually used in this work are missing. It's a bit confusing to mention gradients if they are never used, because it creates mental links with completely different literature. Why self-attention between different weight space features (dimensionalities from 1 to c) are needed?

3. L186 mentions the limitation of Inr2Vec "requires every INR share the same initialization", but in the Experiments this advantage of the proposed method is not analyzed w.r.t. Inr2Vec, so it's unclear if this advantage is really useful. Also, Inr2Vec is not compared in Table 1 for some reason.

4. In Table 1, why there are no results for INR2ARRAY with DWS and no DWS results for CIFAR-10? Since, DWS has source code available I believe this should be straightforward to evaluate and would strengthen the paper if the proposed NFT still dominates. Why INR2ARRAY only improves when combined with NFT and not with NFN?

5. Section 5.1 is confusing, because it mixes methods, experiments and related work together.

6. In Table 3, why NFN can be combined with HNP, while NFT cannot? Using HNP seems important in this experiment, so combining it with NFT could improve the paper.

7. Editing INRs is an interesting experiment, but it is not very clear what would be the realistic use case. The experiment in DWS "Adapting a network to a new domain" seems more useful in this sense.

8. Overall, the experiments seem quite esoteric with unclear real world use cases. For example, the motivation for encoding the weights of INR is not very convincing. This is a general problem with the topic of encoding of neural network weights as researchers seem to have not found really useful and realistic use cases of these models. The experiment "Predicting CNN classifier generalization" tackles a more concrete and useful application, but these also seem tiny networks with a few thousand parameters (Small CNN Zoo in [40]). So overall, this paper would be stronger if some other more practical applications are considered and/or the model is scaled to encoding more realistic networks (like ResNets in the "NeRN - Learning Neural Representations for Neural Networks. ICLR 2023." paper).

9. Some relevant papers/methods are not mentioned/compared in this paper. These include "Self-supervised representation learning on neural network weights for model characteristic prediction. NeurIPs 2021.", which discussed and compared in DWS, and "NeRN - Learning Neural Representations for Neural Networks. ICLR 2023."

**Minor**

The paper needs some polishing to improve the reading experience. Some abbreviations like NFN and SIRENS were used before they are introduced/cited, e.g. L201: " Given a SIREN ...". There are some occasional typos (e.g. L266 "anount") and the hyperrefs are broken (always jump to the first page).

**Questions:**

Please see Questions in Weaknesses above.

**Limitations:**

Limitations are briefly discussed, which is appreciated. The authors could empirically compare time/memory complexity of their method to NFN, DWS and naive self-attention, which could strengthen the paper.

---

> ### Author Rebuttal · Authors · 2023-08-10
>
> Thank you for your review--we appreciate the detailed and thoughtful questions. We will work to disentangle Section 5.1 and improve the clarity of the writing. Here we aim to answer them individually:
>
> > Intuitive explanation of the difference between Eq. 4 and 5 is missing (why the union of sets is used and how is it implemented, by concatenation?) Perhaps, some illustration would help.
>
> We provide (in our PDF response) a diagram illustrating Equations 4 and 5 visually. In Eq 4 the rows of $W^{(i)}$ attend to themselves and the columns of $W^{(i-1)}$. In Eq 5 the columns of $W^{(i)}$ attend to themselves and the rows of $W^{(i+1)}$. But notice that computing Eq 5 for output $i-1$ and Eq 4 for output $i$ amount to a single self attention, as shown in the diagram. Set union (concatenation, as you inferred) is necessary to perform this attention over the rows and columns of different matrices altogether.
>
> > Moreover, it would be very useful to see ablation results with each of the terms (4,5,6) ablated in the model.
>
> We performed this ablation and report the results in our top level reply. We found that the first two terms, which give minimal equivariance, are the important ones, while the last term, which is basically a self attention between features of each weight matrix, is less important.
>
> > Why self-attention between different weight space features (dimensionalities from 1 to c) are needed?
>
> Even if your input is just a single set of weights (c=1), the hidden features of a neural functional such as an NFT always results in more channels (c>1) in the hidden layers. Just as convolutional networks might take in an image with 1 or 3 channels, but 500 channels in its hidden layers. Hence NFT layers need to handle features with arbitrary c.
>
> > L186 mentions the limitation of Inr2Vec "requires every INR share the same initialization", but in the Experiments this advantage of the proposed method is not analyzed w.r.t. Inr2Vec, so it's unclear if this advantage is really useful. Also, Inr2Vec is not compared in Table 1 for some reason.
>
> We will provide an empirical comparison with Inr2Vec on the benchmarks of Table 1. See the top-level reply for initial Inr2Vec results on MNIST, which confirm that Inr2Vec performs poorly on these tasks (19% test accuracy on MNIST). We will also add results for FashionMNIST and CIFAR once they are completed.
>
> > In Table 1, why there are no results for INR2ARRAY with DWS and no DWS results for CIFAR-10? Since, DWS has source code available I believe this should be straightforward to evaluate and would strengthen the paper if the proposed NFT still dominates.
>
> We have added DWS results for CIFAR-10, see the top-level reply for details. The relative performance of DWS to the other methods in CIFAR-10 is consistent with the other datasets (MNIST, FMNIST).
>
> We did not have the bandwidth to implement DWS with Inr2Array in the short response period. We do note that DWS performance tends to be lower than that of NFNs on each benchmark independent of Inr2Array, so it's unclear this would add significant new information over our existing Inr2Array+NFN results.
>
> > Why INR2ARRAY only improves when combined with NFT and not with NFN?
>
> We believe this is because NFT is a more expressive and powerful architecture than NFNs, which was our goal in developing the nonlinear layers that underly NFTs. This makes it better at extracting useful representations from weights for downstream tasks.
>
> > In Table 3, why NFN can be combined with HNP, while NFT cannot?
>
> NFN_HNP and NFN_NP and two different variants whose layers are designed under different symmetry assumptions. In [1], designing layers under NP assumptions lead to layers that are simpler and more computationally efficient, so we designed our NFT layers under NP assumptions for simplicity and tractability purposes.
>
> An HNP variant of the NFT layers would be helpful in some settings and is an interesting direction for future work--though it is likely nontrivial to implement and train scalably in practice.
>
> > Overall, the experiments seem quite esoteric with unclear real world use cases.
>
> We agree that applications such as predicting classifier generalization would be generally useful, though currently they are at small scale. We would like to emphasize that the development of architectures that can process raw weights is very recent, and we believe that our experiments (including the INR tasks) represent promising early steps towards larger scale applications. The scale of our benchmarks in line with that of similar papers published at very recent conferences [2,3], and some of our results (e.g., INR classification) significantly improve performance and results over those works.
>
> > Some relevant papers/methods are not mentioned/compared in this paper.
>
> Thank you for the references, we will cite and discuss those works in our revised paper.
>
> [1] Zhou et al. Permutation Equivariant Neural Functionals.
>
> [2] Navon et al., Equivariant architectures for learning in deep weight spaces, ICML 2023.
>
> [3] Luigi et al., Deep Learning on Implicit Neural Representations of Shapes, ICLR 2023.

---

> > ### Comment · Reviewer_5PwT · 2023-08-14
> >
> > Thank you for the response which partially clarifies some of my concerns.
> >
> > Regarding these Reviewer 73nt questions and responses:
> >
> > 1. > Missing results for NFT without INR2Array for the INR classification task in Table 1.
> >     > We will run this and report the results when they are available.
> >
> > 2. > What is the NFN (and INR2Array^NFN) variant used in Table 1 (NP or HNP)?
> >     > That used NP. We are running an HNP variant for comparison and will include its results in the revised paper
> >
> > Have you been able to obtain the missing results in both 1 and 2?
> >
> > I'm confused by the response in 2, because Table 1 caption says "For the NFN baseline [46] we report the **higher performance out of their NP and HNP variants**", but in the response you are saying it's NP.

---

> > > ### Author Response · Authors · 2023-08-14
> > > **Thanks for reading our reply!**
> > >
> > > Thanks for reading our responses and replying to them. We hope to clarify any remaining questions you have.
> > >
> > > > NFT without INR2Array
> > >
> > > We've finished initial experiments for NFT (no Inr2Array) on MNIST, and present the results here with other Table 1 results for reference. Plain NFTs achieve worse test accuracy than both Inr2Array^NFT, or even plain NFNs. At the same time, the training accuracies shows that plain NFTs are able to fit the data just as well or better than NFNs with much fewer parameters, indicating that it does not lack expressive power but perhaps does not generalize as well.
> > >
> > > The results highlight the importance of Inr2Array: the reconstruction objective and spatially structured latents allow NFTs to learn representations of INR weights that are much more amenable to downstream classification than classification itself. This is analogous to, e.g., self-supervised learning providing benefits for downstream classification in computer vision. We will add this comparison to Table 1 and the discussion of results.
> > >
> > > |                    | MNIST train acc | MNIST test acc | Number of parameters |
> > > |--------------------|-----------------|----------------|----------------------|
> > > | NFN                | 94.9            | 92.9           | 45M                  |
> > > | NFT                | 95.7            | 88.5           | 18M                  |
> > > | Inr2Array^NFT | 100             | 98.5           | 22M                  |
> > >
> > > > Table 1 caption says "For the NFN baseline [46] we report the higher performance out of their NP and HNP variants", but in the response you are saying it's NP.
> > >
> > > Good point--here both statements are correct. We did select the better performing NFN variant to report as baseline, but we found that on each dataset the higher performing NFN variant _was_ NP. This matches the relative ranking of NP and HNP reported in [46, Table 3] itself.

---

> > > > ### Comment · Reviewer_5PwT · 2023-08-16
> > > >
> > > > Thank you for clarifying the concerns and reporting additional results. My major concerns are resolved, so I'm raising the score from 4 to 6.

---

### Official Review · Reviewer_zT4L · 2023-07-06

**Soundness:** 3 good
**Presentation:** 4 excellent
**Contribution:** 3 good
**Rating:** 6
**Confidence:** 4

**Summary:**

This paper introduces a new architecture called Neural Functional Transformer (NFT), that uses an attention-based layer that is equivariant to neuron permutations. NFT is an architecture that is used for tasks where the model is taking an input another neural network's weights (in their case, either an MLP or CNN), which may be e.g. an INR of a some signal or the weights of another trained model. They benchmark their architecture on two INR tasks (image classification, image editing) and predicting classifier generalization from weights.

**Strengths:**

The paper presents the problem of building architectures that operate on other network's weights, which is significant and has useful applications to e.g. working with INRs. The architecture they propose based on attention is original, interesting and clearly presented. The writing of the paper is clear throughout. Their experiments are relevant and explained well.

**Weaknesses:**

Although there are no major weaknesses in this work, I found the empirical performance of NFT slightly underwhelming.
- For CIFAR-10 classification, why isn't there a comparison to Spatial Functa (or usual functa) (which was the inspiration for using spatially structured latent representations)? As shown in Spatial Functa's table 2, they achieve 90% top-1 accuracy, which is significantly higher than 63% achieved by NFT. Both approaches are for image classification using INRs for the images, hence a comparison seems appropriate. Regarding using the same initialization for different INRs, rather than being a weakness, wouldn't that be the go-to approach anyway because it cuts the INR training time by using techniques like MAML to pick the initialization?
- What about direct image classification with NFTs instead of first encoding the INR as an array and then performing classification with that array instead of the original image array?
- For generalization prediction, why isn't there a method corresponding to NFT_HNP in Table 3?

**Questions:**

- I think there's a typo in the caption of Figure 1 in the last sentence.
- The weight spaces handled by the proposed layer are those of CNNs and MLPs. Do you think there are ways to extend this to other commonly used architectures like transformers?
- Neuron permutations come from the fact that MLPs have layers composed like $W_1\sigma(W_2x)$ where $\sigma$ is a point-wise nonlinearity. From this, given a permutation matrix $P$, we get $W_1P^T\sigma(PW_2x) = W_1\sigma(W_2x)$ because $\sigma(PW) = P\sigma(W)$ for any matrix $W$. Is this property satisfied for matrices $P$ that are not permutation matrices? If so, we would also want equivariance to those transformations of the weight space.
- I found Section 3.3 slightly unclear to read, particularly lines 155-158.
- Where is the layer encoding in section 3.4?
- In Section 5.2, do you train two INRs with the same initialization on the the original and edited images and then take their weight difference as the target for the NFT?
- Typo in "amount" in line 266.
- In section 5.3, define the abbreviation HNP for clarify (I think only NP was defined in the beginning of the paper).


**Limitations:**

Adequately addressed.

---

> ### Author Rebuttal · Authors · 2023-08-10
>
> Thank you for your thorough analysis and review. We have updated our paper to correct the minor typos and missing definitions you found.
>
> > What about direct image classification with NFTs instead of first encoding the INR as an array and then performing classification with that array instead of the original image array?
>
> If we understand the question correctly, is the question about why classify INRs at all instead of directly performing the classification in image space?
>
> We share the motivation of [1,2,3] in operating on INR weights since INRs are continuous representations of signals that decouple the memory cost of the representation from the actual spatial resolution. For example, high resolution gridded representations of 3D data can be very expensive and unwieldy to work with.
>
> Also, there would be no need to use NFTs (or any neural functional architecture) in that case since they are designed specifically for handling weight inputs, not images.
>
> Please let us know if your question had a different intended meaning.
>
> > For generalization prediction, why isn't there a method corresponding to NFT_HNP in Table 3?
>
> In this paper, we only define equivariant and invariant attention layers with respect to the Neuron Permutation (NP) group, and not the Hidden Neuron Permutation (HNP). The NP group is simpler to analyze and design layers for, and the finding in [1] is that the resulting layers can be more computationally efficient.
>
> Extending NFTs to HNP would require non-trivial effort and implementation complexity, but could be an interesting direction for future work, especially given that the results of Table 3 do suggest a hypothetical NFT_HNP could have benefits.
>
> > The weight spaces handled by the proposed layer are those of CNNs and MLPs. Do you think there are ways to extend this to other commonly used architectures like transformers?
>
> Yes, we believe that this is another interesting direction for future work. Transformer weight spaces also contain permutation symmetries, not least because they contain feedforward MLP blocks. So it should be feasible to extend NFTs (and neural functionals in general) to create minimally equivariant layers for processing Transformer weights. This extension is not trivial or immediately obvious, since Transformers’ permutation symmetries are more complex than those of feedforward MLPs/CNNs–for example, because they contain residual connections that tie together neurons from different layers and prevent them from being independently permuted. So a first step would be to characterize this more complex symmetry structure, then derive equivariant/invariant layers for it.
>
> > Is this property satisfied for any matrices that are not permutation matrices? If so, we would also want equivariance to those transformations
>
> Yes, for activation functions like ReLU we also get a scaling symmetry, so permutation symmetries would not be the only kind. For a positive scalar $c$ we have $\frac{1}{c}W_2 \sigma(cW_1 x) = W_2 \sigma (W_1 x)$.
>
> For NFTs we focus on permutation symmetries because one is typically dealing with weights produced by an optimization process like SGD, where these scaling symmetries typically do not appear. As [2] found: “SGD’s implicit regularization balances weight norms and, therefore, scale invariance does not seem to play an important role in understanding symmetries of solutions found by SGD.” So there is some evidence from prior literature that accounting for permutation symmetries is sufficient in practice.
>
> > I found Section 3.3 slightly unclear to read, particularly lines 155-158.
>
> We will revise this section to make it clearer. Here we are assuming that the input to the NFT are convolutional weight filters, instead of MLP weight matrices. For a 1D convolution, the filter has shape $\mathbb{R}^{n_i \times n_{i-1} \times k_i}$, where $n_i,n_{i-1}$ are the input and output channels _of the convolution_. Meanwhile, $c$ is the number of channels _of the NFT layer which is operating on the convolution filters_, hence the shape given on L158.
>
> We welcome any additional feedback if this is still unclear.
>
> > Where is the layer encoding in section 3.4?
>
> If the question is referring to the random fourier features (RFF): it is applied right before the first NFT layer, in order to guarantee a consistent channel dimension throughout the NFT.
>
> Suppose the NFT input has a single channel, so each $W^{(i)}\_{jk} \in \mathbb{R}$ is a scalar. Then the RFF independently maps each scalar weight into a vector of length $c$, i.e. it is a function $\mathbb{R} \rightarrow \mathbb{R}^c$.
>
> > In Section 5.2, do you train two INRs with the same initialization on the original and edited images and then take their weight difference as the target for the NFT?
>
> In Section 5.2, the training loss is computed directly in image space, so we never have to train an INR on the edited image at all. To elaborate, the NFT produces edited weights for the INR, and the INR produces a predicted image. The loss is computed between the edited image (target) and the INR’s predicted image. The loss is then backpropagated _through_ the INR back to the NFT that produced the INR’s weights.  We will add this clarification to 5.2.
>
>
> [1] Zhou et al. Permutation Equivariant Neural Functionals.
>
> [2] Navon et al., Equivariant architectures for learning in deep weight spaces, ICML 2023.
>
> [3] Luigi et al., Deep Learning on Implicit Neural Representations of Shapes, ICLR 2023.
>
> [4] Entezari et al. The Role of Permutation Invariance in Linear Mode Connectivity of Neural Networks.

---

> > ### Comment · Reviewer_zT4L · 2023-08-14
> >
> > Thanks for your response! On two of the points above:
> >
> > > If we understand the question correctly, is the question about why classify INRs at all instead of directly performing the classification in image space?
> >
> > I meant that given an image (in a dataset of images), you can fit an MLP to each image (i.e. an MLP that learned the mapping from pixel location to the RGB value) and then directly apply your architecture to the MLPs' weights for image classification, instead of first going from the MLP weights to an array and then applying a model on that.
> >
> > > Yes, for activation functions like ReLU we also get a scaling symmetry.
> >
> > Is that the only other symmetry? I think it would be good to exhaustively classify the possible matrices that will have this property: permutation and scaling are two types, but are they the only types? If there are other types, ideally functional neural networks should be invariant/equivariant to them.

---

> > > ### Author Response · Authors · 2023-08-15
> > >
> > > Thanks for your response!
> > >
> > > > directly apply your architecture [...] instead of first going from the MLP weights to an array
> > >
> > > Good question--the benefit of Inr2Array (encoding the weights into an array trained via reconstruction objective) is that the  reconstruction objective and spatially structured latents allow NFTs to learn representations of INR weights that are much more amenable to downstream classification, resulting in much better performance.
> > >
> > > In response to another reviewer's request, we actually ran NFT directly (without Inr2Array) on MNIST, and present the results here with other Table 1 results for reference. Inr2Array clearly presents a big boost over directly applying our architecture as a classifier. We will add this comparison to Table 1 and the discussion of results.
> > >
> > > |                    | MNIST train acc | MNIST test acc | Number of parameters |
> > > |--------------------|-----------------|----------------|----------------------|
> > > | NFN                | 94.9            | 92.9           | 45M                  |
> > > | NFT                | 95.7            | 88.5           | 18M                  |
> > > | Inr2Array^NFT | 100             | 98.5           | 22M                  |
> > >
> > > > Is that the only other symmetry?
> > >
> > > To our knowledge permutation matrices and positive scaling are the most general symmetry transformations of ReLU networks, without making some assumptions about the value of the weights (minor correction to our earlier response: one can have a distinct positive scalar per neuron). See also Table 1 of [1], which directly tries to answer your question about classifying all possible matrix transformations that leave a network invariant. We'll discuss these results more explicitly in our own work as well.
> > >
> > > We agree that NFs ideally should be equivariant to all symmetries (not just permutations), this would be a challenging and interesting direction for future work.
> > >
> > > [1] Godfrey et al. On the Symmetries of Deep Learning Models and their Internal Representations.

---

> > > > ### Comment · Reviewer_zT4L · 2023-08-17
> > > > **Response**
> > > >
> > > > Thanks for sharing that table. I wonder why NFT doesn't perform as well if applied directly to the INR MLP's weights. The "original" signal that is being turned into an array in Inr2Array is the set of weights of the MLP, so all the information the model needs is there. (Might be useful to run this experiment on the harder CIFAR-10 dataset too in the future.) I saw your response to reviewer 73nt regarding generalization not necessarily being better for a more expressive architecture, but the primary way to motivate a new architecture is indeed via strong empirical performance. I would also encourage including a comparison to Spatial Functa as I mentioned in my original review. Thanks for your responses!

---

### Official Review · Reviewer_e4bE · 2023-07-08

**Soundness:** 3 good
**Presentation:** 3 good
**Contribution:** 2 fair
**Rating:** 6
**Confidence:** 3

**Summary:**

This paper aims to construct neural functional architectures that can handle high-dimensional weight-space objects. Specifically, this paper designs a novel class of weight-space models with neuron permutation symmetries using nonlinear self-attention and cross-attention layers. The proposed method is called neural functional transformers (NFTs). NFTs enable greater expressivity and higher classification accuracy on implicit neural representations (INRs) over existing methods.

**Strengths:**

1. The overall written is good, and the authors provide code for reproduction.

2. The topic of this paper is cutting-edge and the proposed method is novel.

3. The proposed method is simple and its performance for INR classification is good.

**Weaknesses:**

1. It seems straightforward to make self-attention permutation equivariant. What is the actual contribution of section 3.1? Do you compare the performance of original attention and your modified attention?

2. Equation (4) (5) (6) are difficult to read. The authors may represent them via a better way.

3. In table 3, the performance is worse than NFN which is inconsistent with table 1 and table 2. Can you explain the reason of this?

4. Do you evaluate your method on 3D dataset?

**Questions:**

See "Weaknesses".

**Limitations:**

This paper has no potential negative societal impact.

---

> ### Author Rebuttal · Authors · 2023-08-10
>
> Thank you for your suggestions and insightful questions.
>
> > It seems straightforward to make self-attention permutation equivariant. What is the actual contribution of section 3.1? Do you compare the performance of original attention and your modified attention?
>
> The core of our contribution is that naively applying self-attention is permutation equivariant, but not _minimally_ so. If we treat each $W^{(i)}\_{jk} \in \mathbb{R}^c$ as a distinct input vector, then self attention on these $\dim(\mathcal{W})$ inputs would be equivariant to _any_ permutation of the weights, including permutations that are not true symmetries of the weight space (see examples in Figure 1). This means the layer would be overly constrained by symmetries that are not relevant to weight space tasks. On the other hand, the layer we define (Eqs 4-7) is minimally equivariant, which is the core theoretical contribution of Section 3.1.
>
> To confirm the importance of minimal equivariance, we have added ablation experiments where we remove terms from the definition of our equivariant self-attention layer (see top-level reply).
>
> In these ablations _GlobalSA_ is basically a naive application of self attention between features of every weight matrix, and it performs very poorly. This shows that the particular definition of our layer, especially the first two terms which give minimal equivariance, is important for good performance in practice.
>
> > Equation (4) (5) (6) are difficult to read. The authors may represent them via a better way.
>
> We have created an illustration of Eqs 4-6, please see our attached PDF response. This should help illuminate the shapes of the inputs, outputs, and intermediate computations. We will add this figure to our paper.
>
> > In table 3, the performance is worse than NFN which is inconsistent with table 1 and table 2. Can you explain the reason of this?
>
> This is an interesting question, and could be due to a number of factors that separate the small CNN zoo from, e.g., the INR classification tasks. The NFN performance on the small CNN zoo benchmark (table 3) is already quite high (state of the art on the benchmark). Additionally, the size of the training dataset (12,000 training samples) is significantly smaller than for the INR datasets (45,000 training samples, plus data augmentation). So it may be that NFTs only benefit at greater scales and data quantities. We will add this to our paper's discussion of the experimental results.
>
> > Do you evaluate your method on 3D dataset?
>
> Although Inr2Array could be extended to learn a 3D spatial latent encoding each 3D INR, our experiments focused on 2D INR benchmarks. From private correspondence with the authors of [1], it turns out that NFNs already perform quite well on 3D shape INR classification benchmarks, e.g. where each INR encodes a signed distance field for a ShapeNet-10 shape. This is in contrast with 2D INRs where, for example, NFNs classify CIFAR INRs with _much_ lower accuracy than standard CNN baselines do on vanilla CIFAR.
>
> So we focused on 2D INRs since they are, surprisingly, more challenging in the weight space. This aligns with findings from Functa and Spatial Functa [2,3], which also found that classifying 2D INRs was actually more challenging than classifying 3D INRs.
>
> [1] Zhou et al. Permutation Equivariant Neural Functionals.
>
> [2] Dupont et al. From data to functa: Your data point is a function and you can treat it like one.
>
> [3] Bauer et al. Spatial Functa: Scaling Functa to ImageNet Classification and Generation.

---

> > ### Comment · Reviewer_e4bE · 2023-08-18
> > **Update my rating from 5 to 6.**
> >
> > Thank you for your response. All my questions have been addressed. By combining the figure in pdf response and equations, I understand how minimal equivalence is achieved. I see the contributions in section 3.1. Moreover, the authors add ablation experiments which prove the importance of such minimal equivalence.
> >
> > So I update my rating from 5 to 6 and my confidence from 2 to 3.
> >
> > I suggest that the authors describe the weight-space more in Introduction or Preliminaries.

---

### Official Review · Reviewer_BGFR · 2023-07-11

**Soundness:** 3 good
**Presentation:** 3 good
**Contribution:** 3 good
**Rating:** 6
**Confidence:** 3

**Summary:**

The authors consider the principled design of neural functionals, i.e. neural
networks that take other neural networks as inputs. Example tasks include
predicting the generalisation error of a trained network for a given set of
hyperparameters, or trying to reconstruct the images or 3D object encoded in an
implicit neural representation (INR, e.g. Mildenhall et al. arXiv:2003.08934).

The authors propose a transformer-style architecture that consists of a stack of
transformer blocks, each one consisting of a self-attention mechanism followed
by an two-layer fully-connected neural net. The key challenge is to design a
self-attention mechanism that respects the various symmetries of that preserve
the network's behaviour. Naïve self-attention is equivariant to weight
permutations that change network outputs. Previous work (Navon et al. [31] and
Zhou et al. [46]) designed linear layers that are equivariant to neuron
permutation symmetries.

The key contribution of this paper is to propose a particular form of the
self-attention mechanism for neural functionals. It really is a combination of
three attention mechanisms (eq. 4-6), which has however to be approximated in a
rather heavy way: from self-attention between weight-space features to
self-attention between row-column *sums* of the weight matrices. The proposed
mechanism works for both fully-connected layers and convolutional layers.

Given that weight space tasks remain computationally challenging, the authors
also propose an embedding method that maps implicit neural representations into
arrays, an extension of the inr2vec method of De Luigi et
al. [arXiv:2302.05438]. Its performance compares favourably to other methods
when tested on weight-space INR classification of standard image data sets,
where we seek to reconstruct the image encoded in the weights of an INR.

**Strengths:**

- The authors propose a well-designed architecture for a neural functionals.
- The authors show that the self-attention has several desirable properties,
  even in its computationally simplified form.
- The proposed method works well on some tasks chosen by the authors.

As I have no practical experience with neural functionals, I may have missed
some relevant related work or may misjudge the novelty of the proposed solution,
hence my low confidence.

**Weaknesses:**

I believe the article could benefit from some discussion of key concepts regarding neural functionals, for
example the type of tasks in standard benchmarks, to make
the article easier to read on its own.


**Questions:**

The authors introduce a neural functional architecture that is transformer-based
and can deal with fully-connected and convolutional networks. An obvious
question is whether this approach can be extended to transformers? I would
appreciate some discussion of this issue from the authors.


**Limitations:**

The authors discuss computational limitations of the vanilla form of their self-attention layer, and discuss computationally feasible alternatives.

---

> ### Author Rebuttal · Authors · 2023-08-10
>
> Thank you for your feedback and thoughtful review.
>
> > I believe the article could benefit from some discussion of key concepts regarding neural functionals, for example the type of tasks in standard benchmarks, to make the article easier to read on its own.
>
> We will expand the Experiments (Section 5) to elaborate on the types of weight-space tasks one would solve when using neural functionals. We would also be happy to elaborate, both on openreview and in the manuscript, on any other specific topics that were insufficiently explained.
>
> > An obvious question is whether this approach can be extended to transformers?
>
> This is a good idea--Transformer weight spaces also contain permutation symmetries, for example because they contain feedforward MLP blocks. So it should be feasible to extend NFTs (and neural functionals in general) to create minimally equivariant layers for processing Transformer weights.
>
> This extension is not trivial or immediately obvious, since Transformers’ permutation symmetries are more complex than those of feedforward MLPs/CNNs–for example, because they contain residual connections that tie together neurons from different layers and prevent them from being independently permuted. We believe that extending NFTs and neural functionals to non-feedforward architectures is a very useful and important future direction for making this framework more broadly applicable.

---

> > ### Comment · Reviewer_BGFR · 2023-08-14
> > **Thank you for your reply**
> >
> > I have read the author's reply, as well as the other reviews, and I will keep my score.

---

### Author Rebuttal · Authors · 2023-08-10

We thank the reviewers for their thoughtful feedback and questions. Here are some changes and new experiments we have done that will help strengthen this paper:

* Created a diagram illustrating the equivariant self attention (SA) layer (see PDF response)
* Performed ablation experiments on the individual terms in SA to understand the importance of each
* Added empirical Inr2Vec comparison
* Corrected the DWS comparison in Table 1 and added DWS results for CIFAR-10
* Ablated the effect of mixup on the Inr2Array latents in classification tasks (see reply to 73nt)
* Expanded the proof of Theorem 3.1.

## Ablating terms in SA

As suggested, we ran an ablation on the terms in our equivariant self-attention layer. As illustrated by the diagram in our response PDF, the first two terms collectively represent attention between the rows and columns of _adjacent_ weight matrices. The final term is a _global_ attention between features from every layer. Note that in practice we use a tractable variant of the third term described at L133.

For the ablation we either keep only the first two terms (**AdjacentSA**) or only the third term (**GlobalSA**). “FullSA” denotes the original layer.

| Model                      | MNIST test accuracy |
|-----------------------------|----------------------|
| Inr2Array (NFT, FullSA)     | 98.5                 |
| Inr2Array (NFT, AdjacentSA) | 97.6                 |
| INR2Array (NFT, GlobalSA)   | 37.4                 |
| Inr2Vec [1]                 | 19.4                 |

We see that ablating the final term (AdjacentSA) is roughly comparable with the original method, while ablating the first two terms (GlobalSA) drastically harms performance. This is interesting since the first two terms provide minimal equivariance, while the third term alone is not minimally equivariant (but helps propagate information between any two weight matrices). The results emphasize the importance of minimal equivariance.

We will run this ablation on the other benchmarks and will include the results in the revised paper.

## Inr2Vec comparison

We are also running Inr2Vec [1] on the classification tasks (Table 1), as suggested. See the table above for the initial results on MNIST, which empirically confirm that our method outperforms Inr2Vec by a wide margin, and also confirms the limitations of Inr2Vec on settings without shared initialization.

We are running this comparison for the other datasets (Fashion MNIST, CIFAR) and will include the results in the revised paper.

## Corrected DWS comparison (+ CIFAR10)
As pointed out by 73nt, the results for DWS [2] in Table 1 are those reported by [2] itself, but INR datasets were constructed differently than the ones we use, which were produced by [3]. We have corrected this issue by running DWS ourselves on the latter datasets, and present the results below. This also adds DWS results for CIFAR-10, which were missing before. With the corrected numbers, the relative performance of different methods in Table 1 remains unchanged.

| Test accuracy  | MNIST | FashionMNIST | CIFAR |
|----------------|-------|--------------|-------|
| DWS (32 channel)| 74.7 | 67.5         | 42.3  |
| DWS (512 channel)| 61.6 | 62.0        | 42.9  |
| Inr2Array^NFT (ours) | 98.5 | 79.3        | 63.4  |

**Baseline details**: We trained the DWSNets of two different widths (width 32 and 512), with the same data augmentation scheme as for the other methods. For each dataset and channel size we swept learning rates in $[1e-3, 5e-3, 1e-4, 5e-4]$, following the protocol in [2].

## Expanded proof of Thm 3.1
Reviewer 73nt pointed out that the proof of equivariance of SA could be made more detailed. We provide that here.

**Lemma**: attention (Eq 3) is equivariant to permutations $\pi \in S_d$ to each
key/value/query input. In particular, let $[\pi q][i] = q[\pi^{-1}(i)]$ permute the entries
of the input vectors. Then:

$$Attn\left(\pi q, \left\\{\pi k\_p, \pi v\_p\right\\}\_{p=1}^N\right)[i]
= \left[\sum_p \left(\frac{\exp(q \cdot k_p)}{\sum\_{p'} \exp(q \cdot k\_{p'})}\right) (\pi v_p)\right][i]
= \sum_p \left(\frac{\exp(q \cdot k_p)}{\sum\_{p'} \exp(q \cdot k\_{p'})}\right) v_p[\sigma^{-1}(i)]
= Attn\left(q, \left\\{k\_p, v\_p\right\\}_{p=1}^N\right)[\pi^{-1}(i)]$$

Note that in the first equality we use $\pi q \cdot \pi k = q \cdot k$.

**Equivariance of Eqs 4-7**: As the KQV projections are only linear projections along the channel dimension,
$K(\sigma W) =\sigma K(W)$, where $K$ denotes all keys produced
by linearly projection of $W$ by $\theta_K$ (and similarly for $Q,V$). Then consider,
for example, how the first term (Eq 4) change under $W \mapsto \sigma W$:

$$Attn\left(\[\sigma Q\]^{(i)}\_{j,:}, \left\\{ (\sigma K, \sigma V)^{(i-1)}\_{:,q} \right\\}\_q \bigcup \left\\{ (\sigma K, \sigma V)^{(i)}\_{p,:}\right\\}\_p\right)\[k\] = Attn\left( \sigma\_{i-1} Q\_{\sigma\_i^{-1}(j),:}, \left\\{ (\sigma\_{i-1} K^{(i-1)}\_{:,q}, \sigma\_{i-1} V^{(i-1)}\_{:,q}) \right\\}\_q \bigcup \left\\{ (\sigma\_{i-1} K^{(i)}\_{p,:}, \sigma\_{i-1} V^{(i)}\_{p,:}) \right\\}\_p \right)\[k\]$$

Note that permutations along the KV set indices are ignored since set order is irrelevant. Applying the lemma, we have:

$$Attn\left(Q^{(i)}\_{\sigma\_i^{-1}(j),:}, \left\\{ (K, V)^{(i-1)}\_{:,q} \right\\}\_q \bigcup \left\\{ (K,V)^{(i)}\_{p,:}\right\\}\_p\right)\[\sigma\_{i-1}^{-1}(k)\]$$

Notice that compared to Eq 4 the output indices have changed $j \mapsto \sigma\_i^{-1}(j)$ and $k \mapsto \sigma\_{i-1}^{-1}(k)$.

We can similarly show equivariance for the second term (Eq 5), and the third term (Eq 6) is equivariant to any permutation since it is treating each weight as a token. We omit the full details due to character limit, but will add them to the paper.

## References

[1] De Luigi et al. Deep Learning on Implicit Neural Representations of Shapes.

[2] Navon et al. Equivariant Architectures for Learning in Deep Weight Spaces.

[3] Zhou et al. Permutation equivariant neural functionals.

---

### Decision · Program_Chairs · 2023-09-21

**Decision:**

Accept (poster)

**Comment:**

The paper suggests a novel transformer-like architecture that can equivariantly process the weights of other neural networks, building on recent works that suggested linear layers for that setup. In addition, the authors propose INR2array - a novel approach for computing invariant representations for weights of implicit neural representations.  The reviewers found this extension to be natural and well-designed and appreciated the experimental results. On the other hand, reviewers were concerned by some cases in which the architecture performs worse than previous linear architectures, by the lack of experiments on 3D data, lack of clarity, and some missing experiments and baselines and unfair evaluation The authors posted a rebuttal that addressed some of these concerns.

All reviewers finally converged to an accept decision.